# Unraveling the molecular basis of substrate specificity and halogen activation in vanadium-dependent haloperoxidases

P. Zeides[1,2,7], K. Bellmann-Sickert[2,7], Ru Zhang[2,3], C. J. Seel[1], V. Most [4], C. T. Schoeder[4], M. Groll [5] & T. Gulder [1,2,3,6] ✉

Vanadium-dependent haloperoxidases (VHPOs) are biotechnologically valuable and operationally versatile biocatalysts. VHPOs share remarkable active-site structural similarities yet display variable reactivity and selectivity. The factors dictating substrate specificity and, thus, a general understanding of VHPO reaction control still need to be discovered. This work's strategic single-point mutation in the cyanobacterial bromoperoxidase *Am*VHPO facilitates a selectivity switch to allow aryl chlorination. This mutation induces loop formation that interacts with the neighboring protein monomer, creating a tunnel to the active sites. Structural analysis of the substrate-R425S-mutant complex reveals a substrate-binding site at the interface of two adjacent units. There, residues Glu139 and Phe401 interact with arenes, extending the substrate residence time close to the vanadate cofactor and stabilizing intermediates. Our findings validate the long-debated existence of direct substrate binding and provide a detailed VHPO mechanistic understanding. This work will pave the way for a broader application of VHPOs in diverse chemical processes.

Halogenation plays a crucial role in molecule construction and manufacturing, as halogenated compounds are found in almost all areas of our society, ranging from solvents, refrigerants, propellants, plastics, pesticides to drugs. For example, in annual sales of the 100 most popular pharmaceuticals, 13% of active pharmaceutical ingredients (APIs) contain chlorine or bromine. Even more impressive, 63% of these blockbuster drugs require a halogenation step within their manufacturing process[1]. The broad versatility of organohalides stems from their high reactivity and chemical orthogonality, allowing for various selective transformations, particularly cross-coupling and substitution chemistry. Generating organohalides still mainly involves adding corrosive molecular halogens in combination with metal catalysts or electrophilic organohalogen reagents that enable nucleophilic substitution ($S_N2$) or electrophilic aromatic substitution ($S_EAr$)[2]. However, these reactions pose challenges regarding their environmental impact, sustainability, and selectivity, mainly due to the toxic, corrosive, and non-atom economic nature of the reagents employed and their tendency to produce complex product mixtures. Although recent achievements in homogeneous catalyst development have led to more selective catalytic halogenation approaches, these catalysts tend to be intricate, expensive, and time-consuming to synthesize.

In contrast, nature has evolved different strategies to create *C,X*-bonds precisely under mild conditions, as evidenced by ca. 5000 halogenated natural products isolated to date[3–8]. The most common strategy in nature is electrophilic halogenation[9–12] facilitated by vanadium- (VHPOs) and heme-dependent haloperoxidases, and

[1]Biomimetic Catalysis, Catalysis Research Center, TUM School of Natural Sciences, Technical University of Munich, Garching, Germany. [2]Faculty of Chemistry and Mineralogy, Institute of Organic Chemistry, Leipzig University, Leipzig, Germany. [3]Organic Chemistry, Saarland University, Saarbruecken, Germany. [4]Faculty of Medicine, Institute for Drug Discovery, Leipzig University, Leipzig, Germany. [5]Department of Bioscience, Center for Protein Assemblies, TUM School of Natural Sciences, Technical University of Munich, Garching, Germany. [6]Synthesis of Natural-Product Derived Drugs, Helmholtz Institute for Pharmaceutical Research Saarland (HIPS), Helmholtz Centre for Infection Research (HZI), Saarbrücken, Germany. [7]These authors contributed equally: P. Zeides, K. Bellmann-Sickert. ✉e-mail: tanja.gulder@uni-saarland.de

flavin-dependent (FAD) halogenases. VHPOs exhibit an attractive biotechnological potential for industrial applications[13–17] because of their unusually high stability, high tolerance to synthetic reaction conditions, such as organic solvents, and their broad substrate range[18–21]. In addition, they are capable of oxidizing halides (I⁻, Br⁻, Cl⁻) in the presence of hydrogen peroxide rather than utilizing complicated electron delivery chains, e.g., via nicotinamide cofactors (Fig. 1)[22]. VHPOs are categorized according to the most electronegative halide they can oxidize, leading to chloroperoxidases (VCPOs), bromoperoxidases (VBPOs), and iodoperoxidases (VIPOs). The general mechanism for VHPO-catalyzed reactions is based on the two-electron oxidation of halides[9,10,23]. Theoretical and experimental studies propose that this process starts with hydrogen peroxide coordinating to the vanadium(V) center in **1**, forming a peroxo-vanadate complex (Fig. 1a).

Subsequent attack of a halide ion at the partially positively charged oxygen atom in **2** leads to an electrophilic halogen species, most likely in the form of diffusible hypohalous acid (HOX), which reacts with a suitable substrate[9]. The vanadium(V) oxidation state is retained in the catalytic cycle. Therefore, the metal cofactor remains redox-neutral, and consequently, no regeneration is required. The existence of a binding site for organic molecules to be halogenated in VHPOs is controversially discussed[24]. In several reports, a substrate binding site for organic substrates has been proposed[24,25] because of the low halogenation reactivity of HOX in bulk solution and the selective halogen delivery catalyzed by VCPOs from *Streptomyces* bacteria[26–34], but its existence has yet to be proven.

Superimposition of X-ray structures of VBPOs from different species (from *A. marina*, *C. pilulifera*, *C. officinalis*, and *A. nodosum*)

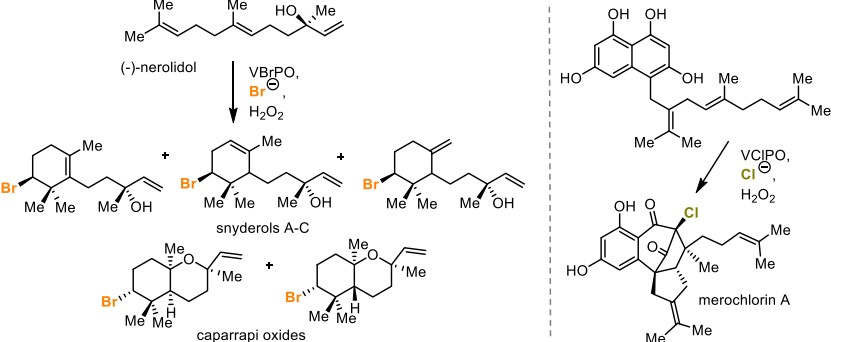

**a) Schematic VHPO halogenation**

**b) Highly conserved active sites in VBPOs and VCPOs**

**c) Non-selective and Selective Halogenations catalyzed by VHPOs**

(-)-nerolidol

VBrPO, Br⁻, H₂O₂

snyderols A-C

caparrapi oxides

VCIPO, Cl⁻, H₂O₂

merochlorin A

**d) This work: Evolving the VHPO from *A. marina* to explore the structure-function relationship of VHPO-catalyzed halogenations**

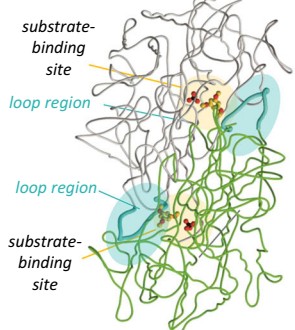

substrate-binding site

loop region

loop region

substrate-binding site

Computer-guided, single-point mutation induced a structurally defined **loop** interacting with the neighboring subunit. This interaction formed a **tunnel** leading to the vanadate cofactor containing the active site.

→ Structuring of a substrate binding site

→ Altering the halogen specificity: chlorination of aromatic and dicarbonyl compounds possible

**Fig. 1 | Central questions on vanadium-dependent haloperoxidase (VHPO) mechanism. a** Schematic representation of the mechanism of the VHPO-catalyzed halogenation. **b** Structure superimposition of active site residues (green) and residues proximal to the active site (gray) of selected vanadium-dependent haloperoxidases from *A. marina* (PDB: 5LPC), *A. nodosum* (PDB: 1QI9), *C. officinalis* (PDB: 1QHB), *C. pulifera* (PDB: 1UP8), *C. inaequalis* (PDB: 1IDQ). The residue nomenclature follows the sequence of the bromoperoxidase of *A. marina*. The backbone in gray is derived from the crystal structure of *A. marina* (PDB: 5LPC). **c** Selected example of a non-selective and a selective halogenation catalyzed by algal VHPOs[38,50] and Streptomyces CNH-189, respectively[30]. **d** This work delivers molecular insights gained by single point mutations of R425 and the structure-activity analysis of this mutant.

with the VCPO from *C. inaequalis* revealed that the key catalytic residues in the active site are superimposable despite overall low sequence similarity between VBPOs and VCPOs (e.g., *An*VHPO vs *Ci*VHPO; 21.5% sequence identity, see SI, SFig. 21)[21,35]. In all VHPOs known so far, the vanadate anion is bound to an axial histidine and further stabilized by hydrogen bonding interactions with the protein backbone. The amino acids involved in vanadate coordination and cofactor binding are conserved in position and orientation in this class of enzymes (Fig. 1b). Nevertheless, VHPOs differ in their capability of oxidizing different halogens. The halogen specificity is thus due to structural changes in an outer sphere surrounding the active center. Despite intense mechanistic and structural studies, especially on HOX generation, essential halide, and substrate specificity questions remain unresolved.

Here, we set out to decipher the structural factors determining halogen specificity in VHPOs. Using computer-assisted protein design[36], sampled mutations allowing for sequence variability to influence chlorination rate while maintaining structural integrity and stability were predicted in the cyanobacterial VBPO from *Acaryochloris marina*. The Arg at position 425, located outside the active site, stood out as being replaceable with smaller and polar amino acids, such as Ser (R425S). Mutagenesis experiments in the second shell combined with activity screening assays revealed that R425S-VCPO was transformed into a chloroperoxidase capable of efficient aromatic chlorination, even on a preparative scale. Subsequent X-ray crystal structure analysis of the R425S variant visualized the formation of structural elements (residues 390–404) that are intrinsically distorted in wild-type *Am*VHPO. Intriguingly, the structural motif is only involved in intermolecular interactions with the active site of the neighboring subunit. There, it introduces a tunnel to the catalytic metal center and, at the same time, forms a precise binding site ideal for aromatic substrates. Thus, the combined computational, structural, and chemical approach revealed that the halogen specificity is coordinated at the intersection of subunits, and an induced fit mechanism for substrate binding is proven by co-crystallization experiments of trimethoxybenzene (**7**) with the R425S-VCPO. In summary, the validated substrate binding channel and the halogen specificity are significant observations for enzyme engineering and database-driven predictions of VCPO-enzyme functions.

## Results

### Computational screening for mutation sites close to the vanadate binding site

For elucidating the halogen specificity in VHPOs, we chose the vanadium-dependent haloperoxidase *Am*VHPO from the cyanobacterium *Acaryochloris marina* MBIC 11017[21]. *Am*VHPO is available in high yields (30 mg L$^{-1}$) using a recombinant *E. coli* expression system and is structurally well characterized[21]. In addition, the enzyme shows remarkable robustness towards organic solvents and heat, together with a broad substrate scope for aromatic bromination, making it a perfect candidate for mechanistic investigations and biotechnological applications[21,22,37–40]. We started our studies by identifying amino acids near the active site that can be modified without affecting the expression of the enzyme (Fig. 2a). Using a model of the *Am*VHPO structure (cf. SI, SFigs. 13 and 14)[21], we designed the sequence using the deep learning-based method ProteinMPNN[41], including positions that are no further than 10 Å away from K428 and show a solvent-accessible surface area (SASA) > 0 (for details see SI, chapter 7.2). Interestingly, position R425 was predicted to respond well to substitutions with smaller amino acids, such as Ala or Ser (Fig. 2b). Notably, this arginine residue is located three amino acids away from the active site lysine (Lys428 *Am*VHPO, see SI, SFig. 15b). While the sequence alignment with other VHPOs revealed R425 in VBPOs from the red algae *Corallina officinalis* and *C. pilulifera*, chlorinating VHPOs from the fungus *Curvularia inaequalis* and the brown alga *Ascophyllum nodosum* consist of

a Trp in these positions (Fig. 1b). Indeed, Izumi et al.[42] showed that a mutation of R397 in *Cp*VHPO (corresponds to R425 in *Am*VHPO) to Phe or Trp increased chlorination activity. Additional calculations using the cartesian ΔΔG protocol implemented in Rosetta[43,44] revealed that substitutions of R425 by small side chains do not confer an energetic advantage (see SI, chapter 7.4 for details). However, these mutations increase solvent accessibility at the enzymatic site (see SI, SFigs. 15c and 18).

### Changing *Am*VHPO reactivity by mutagenesis at R425

We created a library of mutant enzymes by site-saturated mutagenesis of the targeted R425 to validate the calculated results and screened the catalytic fidelity of the obtained mutants using the UV monochlorodimedone (MCD, **3**) assay[45,46]. The results clearly showed that position 425 is critical for the substrate specificity of *Am*VHPO (see SI, chapter 3). Strikingly, the conversion of **3** by bromination and chlorination drastically increased from 53% for bromination and 8% for chlorination for the wild-type *Am*VHPO to 91% and 74%, respectively, for the R425S variant (Fig. 3 and SI). The exchange of R425 by Ala (21% bromination of **3**; 28% chlorination of **3**) and Thr (83% bromination of **3**; 11% chlorination of **3**), which were both likewise predicted by ProteinMPNN (cf. Fig. 2), also showed an overall enhanced halogenation activity compared to the wild-type, but this was not as significant in terms of chlorination as observed for the R425S mutant. Interestingly, substituting R425 with either Phe (2% chlorination of **3**) or Trp (10% chlorination of **3**) displayed almost no change or even a decrease in chlorination activity.

Enzyme kinetics for chloride, bromide, and hydrogen peroxide as substrates were determined using the MCD assay with saturating levels of the respective remaining substrates at the optimum pH value (pH 6.0) to classify the reaction as pseudo-first order (see SI, chapter 5). Kinetic parameters for chloride and H$_2$O$_2$ were determined based on Michaelis-Menten curves. The $K_M$ values for H$_2$O$_2$ were similar between the R425S mutant (66 μM) and the wild-type (60 μM)[21], while the binding constants for chloride differed significantly. Only for the mutant, it was possible to determine the $K_M$ for chloride, leading to 167 mM, which is in line with that of other VHPOs showing significant chlorinating ability, such as the *An*VHPO (344 mM) from brown algal[42] or the VCPO from the deep-sea hydrothermal vent fungus *Hortaea werneckii* (237 mM)[47]. Taken together, the change of the i - 3 amino acid did not affect the binding of the oxidant H$_2$O$_2$ to the enzyme's active site but dramatically affected chloride processing. As different amino acids three positions apart from the active site lysine (Lys428) trigger an enhanced chlorination activity in different VHPOs, such as, e.g., Phe and Trp in *Cp*VHPO[42], Trp in *An*VHPO (see PDB: 1QI9) or Ser in *Am*VHPO, the halogen specificity cannot be traced back to a direct interaction of a single amino acid residue with the halogen or the vanadate cofactor. Different amino acids influence the oxidation potential towards chloride and/or accelerate the speed of oxidation in the class of VHPO enzymes. Thus, a more systemic analysis of VHPOs is needed to reveal the structure-activity relationship.

Additionally, alternative substrates were examined for enzymatic halogenations. The R425S mutant preferentially accepts aryl derivatives and thus exhibits a similar substrate scope as the wild-type enzyme. A comparison of our standard reaction with phenol red (**5**) revealed that the R425S variant chlorinated both substrates. At the same time, no activity was measured in the presence of the wild-type enzyme or the R425D mutant (Fig. 3b).

### Molecular insights into *Am*VHPO variants

To understand the role of position 425 for halogen specificity in *Am*VHPO, we crystallized each, a catalytically highly competent VCPO (R425S, PDB ID 8Q21 and 8Q22; see SI, STable 5) and a VBPO (R425D, PDB ID 8Q20; see SI, STable 5) mutant by the hanging drop vapor diffusion method. An isosteric phosphate, obtained from the reservoir

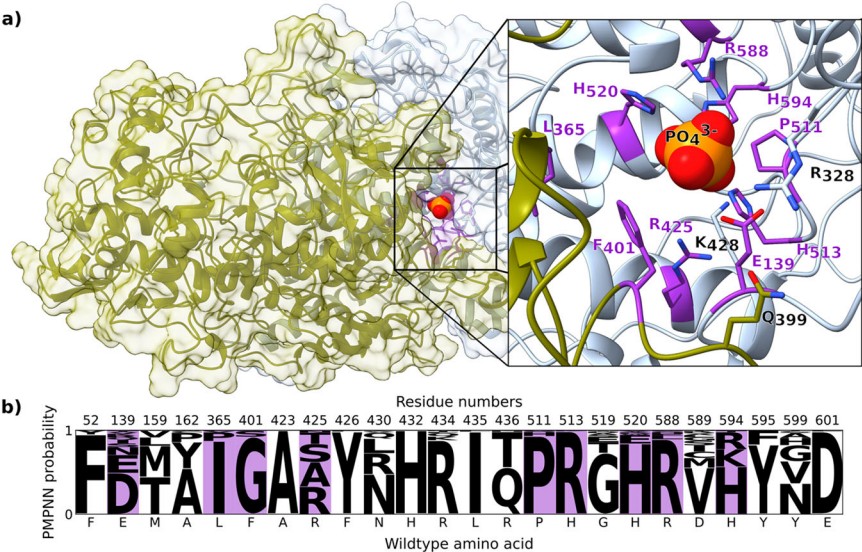

**Fig. 2 | Sequence design of residues in proximity to the active site.**
**a** Visualization of the AmVHPO model. Two chains of the decamer are shown. The design of surface-exposed residues was simulated in a 10 Å radius of the catalytic residue K428. Designed residues pointing towards the substrate channel are highlighted in violet. The phosphate position was superimposed from 5LPC[21].
**b** Conditional probabilities predicted by ProteinMPNN[41] for the designed positions in the AmVHPO dodecamer model.

solution[21], replaced the cofactor vanadate in the enzyme crystals. Both mutant structures feature the identical characteristic assembly of a hexamer of dimers, arranged in a "head-to-tail" orientation with superimposable key catalytic residues when compared to the wild-type (Fig. 4a). Remarkably, variant R425S showed an interaction between Ser425 and a region of the adjacent subunit (Fig. 4b, highlighted in blue). In detail, residues 390–404 in the wild-type structure lack a defined electron density due to flexibility. In contrast, in R425S, this section consolidated in a loop structure, strongly influenced by the introduced serine residue (Fig. 4b, highlighted in blue). The hypothesis that this loop is responsible for the halogen specificity is supported by the crystal structure analyses of the AmVHPO mutant R425D (cf. Fig. 5), which has no chlorination activity and lacks a structured conformation of residues 400–405.

A long-standing hypothesis is that VHPO-catalyzed halogenation of organic compounds occurs outside the enzyme, with the corresponding hypohalous acid diffusing freely in solution. However, there is increasing evidence for a more complex mechanism within the catalytic center of these enzymes[24,48,49]. Recently, extensive investigations on the bacterial VCPOs operating with high selectivity in the napyradiomycin and merochlorin biosynthesis[33] and computational studies together with kinetic experiments on CiVHPO[25] hinted at a substrate binding site in these VHPOs. This assumption aligns with earlier investigations demonstrating that indoles and terpenes are preferentially brominated over MCD (**3**) when these substrates are present in equimolar concentrations[25,50]. Despite these important studies, there is still a massive debate on the existence of a substrate-binding site in VHPOs and whether the halogenation occurs within the substrate enzyme complex or outside the enzyme by freely diffusing HOX. Competition assays for the chlorination of **3** in the presence of varying amounts of TMB (**7**) showed a preference for TMB over **3** (see SI, chapter 6), thus supporting our hypothesis of a substrate binding site in R425S-AmVHPOs. Encouraged by this result, we started co-crystallization experiments using **7** as a ligand. TMB (**7**) only binds to the AmVHPO once the ordered loop region at the interface of two enzyme subunits (loop region, residue 390–404, cyan, Fig. 4c) is defined. In the R425D mutant, this motif is only partially present (residue

390–400), so no interactions with **7** can occur. These molecular findings agree with the wild-type structure, in which residues 390–404 are also flexible. Thus, the plasticity of the specificity pocket depends on the introduced mutant, and only a small modular sequence motif coordinates substrate selection (see SI SFig 21). No co-crystal structure of **7** and the R425D or the wild-type enzyme could be obtained. The effect of the R425S mutation on loop flexibility and pocket formation was further confirmed by 1.6 μs of Hamiltonian replica-exchange molecular dynamics (MD) simulations for each system[51]. In this enhanced sampling method, multiple replicas of the same system are performed at different temperatures and exchange their state at regular intervals with a defined probability. High energy barriers are more easily crossed which allows for extensive conformational sampling of small protein domains in an acceptable simulation time. In the performed simulations, the loop showed a distinctly different conformational profile (see SI, SFig. 19A) in the wild-type and R425S mutant. In agreement with the crystal structure, we observed higher flexibility for several positions in the loop of the wild-type. In particular, position Phe401 stood out as less stabilized as compared to the wild-type protein (see SI, SFig. 19B). As this residue is close to the active site and identified as part of the binding pocket, a certain set of side-chain orientations may be required to allow TMB access and binding (see SI, SFig. 19D). This may indicate a conformationally dependent mode of substrate access or binding.

Figures 5a and 5b depict the protein residues surrounding TMB (**7**) in the substrate-R425S-AmVHPO complex. These residues include Glu139 with Gln399 and Phe401 from the neighboring subunit in proximity to the vanadate cofactor (replaced by phosphate in the X-ray crystal structure). These amino acid side chains favor the anchoring of aromatic substrates. A distance of 3.6 Å between Phe401 and TMB (**7**) indicates a strong π-π stacking that stabilizes the aryl-moiety of **7**. Hydrogen bonding of Glu139 and Gln399 (2.8 Å) and their interaction with the active site His513 facilitate the electrophilic aromatic chlorination (S$_E$Ar). Taken together, the combination of a narrowly shaped specific binding pocket that extends the residence time of substrates near the active site, together with structural rearrangements forging a tunnel structure at the interface of two neighboring subunits, enables

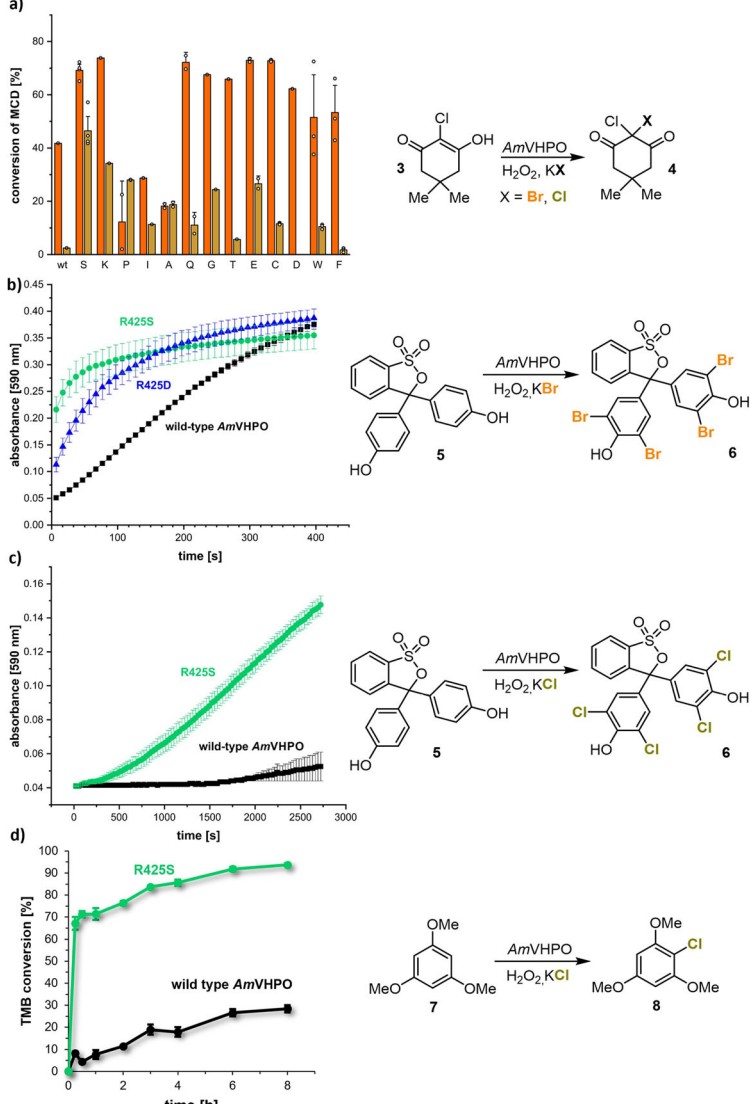

**Fig. 3 | Mutant screening and aromatic halogenations of MCD (3) and phenol red (5). a** Screening of bromination (orange) and chlorination (ocher) activities of relevant R425 mutants using monochlorodimedone (MCD, **3**) assay. The serine variant stands out with a ten times higher conversion of MCD (**3**) than the wild-type for the chlorination reaction. Comparison of **b** the aromatic bromination and **c** aromatic chlorination ability over time using phenol red (**5**) and **d** 1,3,5-trimethoxybenzene (**7**, TMB) using the wild-type *Am*VHPO and the variants R425D and R425S. Data in **b**), **c**) and **d**) are depicted as mean ± SD from 3 independent experiments performed in technical triplicates. Source data are provided as a Source Data file; *Am*VHPO = vanadium-dependent haloperoxidase from *Acaryochloris marina*.

chlorination reactions in our engineered R425S-*Am*VHPO variant. Similarly, in our MD simulations, Phe401 showed a distinctly different conformational space for the R425S (see SI, SFig. 19). These results are consistent with findings in FAD halogenases in which tunnel formation can abolish HOX leakage, leading to increased halogenation efficiency[52].

To shed light on the effects of Glu139 and Phe401 on the catalytic chlorination process, we replaced each of these two amino acids with glycine, generating conformationally more flexible double mutants R425S E139G and R425S F401G, respectively. Intriguingly, the chlorination reaction no longer took place in the F401G mutant, as shown in the enzymatic activity and halide specificity assays (see SI, SFig. 11), but was just slowed down for brominations (cf. SI, SFig. 10). This hints at the binding of the aryl substrate being decisive for chlorinations but only having minor effects on bromination reactions and is consistent with the increased stability of the aryl binding pocket observed in the MD simulations. The E139G variant, however, showed a different behavior. A significant decrease in the chlorination rate of MCD (**3**,

conversion of **3** was decreased by 20% after 15 min) and phenol red assay (**5**) was observed (see SI, chapter 4). Moreover, the chlorination of TMB (**7**) was only detectable in traces according to GC analysis. This emphasizes the supporting function of the carboxylic acid for delivering the electrophilic halide species HOX to the substrate and its role in stabilizing the cationic Wheland intermediate **9** (Fig. 5c).

A small range of other aromatic substrates was successfully transformed to the chlorinated products **10**-**15** employing the R425S mutant (Fig. 5d), showing that the observed chlorination activity introduced by the single-point mutation is not restricted to **7**. Only chlorinated TMB **8** was formed using the wild-type *Am*VHPO albeit in low quantities (<10%). Larger molecules, however, such as carbazoles or stilbenes (cf. SI, chapter s11.9 and 11.10), were not accepted by R425S-*Am*VHPO but served as substrates for wild-type *Am*VHPO catalyzed brominations[21,37]. Higher chemical yields were achieved in enzymatic chlorinations (except for **15**) when compared to those accomplished using chemical reagents (e.g., NCS with 1,4-diazabicyclo[2.2.2]octane (DABCO)). In addition, excellent regioselectivities

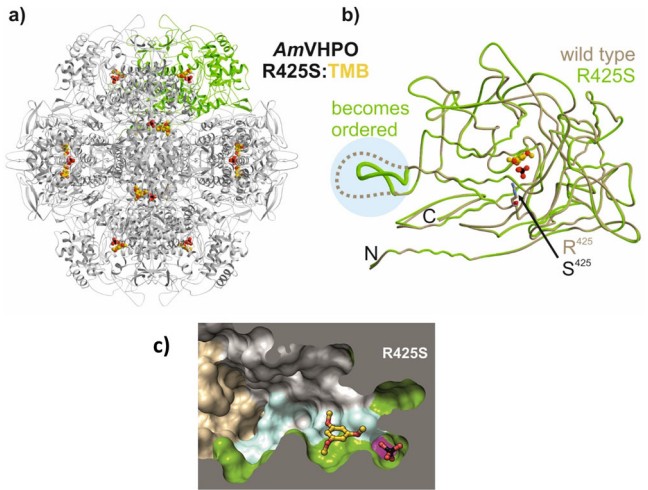

**Fig. 4 | X-ray structure of the AmVHPO-R425S mutant in complex with 1,3,5-trimethoxybenzene (7, TMB). a** Ribbon diagram of dodecameric AmVHPO-R425S mutant with its surrogate TMB (**7**, one subunit is depicted in green, **7** and phosphate (PO$_4^{3-}$) are shown as a ball-and-stick model with gold carbon atoms; PDB ID 8Q22). **b** Structural superposition of AmVHPO-R425S (green) and wild-type AmVHPO (tan, PDB ID 5LPC). Dots indicate a loop region that lacks defined electron density in the wild-type structure (residues 390–404, highlighted in cyan) but which adopts a defined motif in the mutant. **c** Surface cross-section of the AmVHPO-R425S variant in complex with **7**. The cartoon represents one of 12 active sites in AmVHPO. The substrate binding pocket comprises two AmVHPO subunits shown in green and gray, respectively. Other subunits of the dodecamer are colored brown. Residue 425 (magenta) has a significant impact on the shape of the specificity pocket: TMB (**7**) is stabilized by a defined loop region from the adjacent subunit (residues 390–404, highlighted in cyan), which is fully resolved in R425S. AmVHPO = vanadium-dependent haloperoxidase from *Acaryochloris marina*.

were detected for the enzymatic transformations, which significantly exceeded the intrinsic selectivities, further corroborating the existence of a substrate binding site in R425S-AmVHPO.

## Discussion

Vanadium-dependent haloperoxidases (VHPOs) are, in principle, ideal enzymes for applications in industrial processes as they are robust to organic synthetic reaction conditions and need just simple halides together with H$_2$O$_2$ and vanadate to be operative. The structural features responsible for halogen specificity are still elusive as the amino acids inside the active site in VHPOs are highly conserved. Therefore, halide specificity must be controlled around the vanadate binding site in the second or third sphere. The majority of identified VHPOs produce short-lived, highly reactive hypohalites, which exhibit non-specific halogenation reactivity towards diverse substrates[9,53]. In contrast, highly stereoselective chlorofunctionalizations within the biosynthesis of napyradiomycins and merochlorins (cf. Figure 1c) by VHPOs from marine *Streptomycetes* are known[26]. This led to a highly controversial discussion on the location of halogenation, inside or outside the enzyme.

Our study reveals that position 425 in AmVHPO, located three amino acids aside from the active site Lys428, plays an important role in halogen specificity and substrate binding. Exchanging Arg425 by serine organizes residues 390–404 in R425S to a defined loop. This structural variation has no immediate influence on its protein subunit but causes mutual interactions of two neighboring monomers in the 'head-to-tail´ orientation of the dodecameric enzyme structure. Consequently, the loop triggers the formation of a defined tunnel with the vanadate cofactor at the end. Within this tunnel, a combination of structure-induced interactions is observed to be responsible for substrate halogenation. A compelling interplay of Glu139, His513, and

Gln399 from the neighboring subunit near the active site was induced in the R425S-variant (Fig. 6). Although the key catalytic residues in the first coordination sphere of cofactor binding are still strictly conserved within the R425S-AmVHPO mutant, the Ser425 from the adjacent subunit alters the hydrogen-bonding network within the active site, thus forming a tunnel protecting the vanadate in the active site. This might reduce the access of solvent molecules to the vanadate cofactor and thus most likely abolish the degradation of the reactive Cl$^+$ species. In addition, this tunnel might also be responsible for the observed regioselectivity and substrate specificity due to the defined space accessible in the R425S mutant (see Fig. 5d). Such active sites with hampered accessibility in VHPOs have been reported only in the structurally characterized bacterial VCPOs, catalyzing stereoselective chlorofunctionalizations[33], and the VHPO from *Zobellia galactanivorans* (ZgVHPO)[54]. Nevertheless, comparing the structures of the wild-type VBPOs from *A. marina*, *C. officinalis*, and *C. pulifera* with those of VHPOs exhibiting chlorination activity (*A. nodosum, C. inaequalis, Streptomyces* CHN-189) given the knowledge gained in the presented study a significant correlation between the exposure of the vanadate cofactor in their x-ray crystal structures (Fig. 6) and their halogenation activity becomes obvious (for details see SI, chapter 9, SFig. 18). Exchanging the highly conserved Arg (position 425 in AmVHPO) in the VBPOs from *C. officinalis* (position 396), and *C. pulifera* (position 397) by a Ser resulted likewise in a significantly enhanced chlorination reactivity. This further supports the conclusions drawn from our mutagenesis experiments. While the active site vanadate is located at the end of a broad funnel in all VBPOs, ensuring fast substrate access and release of the electrophilic HOX species, different tunnel structures toward the prosthetic group are visible in all x-ray structures of chlorinating VHPOs. In addition, a direct correlation between halogenation activity and the extent of the tunnel is visual—the more shielded the cofactor, the higher and more specific the chlorination activity. The same structure-reactivity relationship is visible when comparing the structures of the wild-type AmVHPO with the R425D, and R425S mutants (cf. SI, chapter 9, SFig. 24). The access to the active site close to the interface between two protein subunits becomes drastically reduced in the R425S mutant compared to R425D and the native form of the enzyme.

The interaction between those monomers is primarily determined by loop structures (cf. SI, chapter 9, SFig. 23). While in the wild-type AmVHPO, the upper loop is not resolved, leaving broad access to the vanadate group, two distinct loops in CoVHPO and CpVHPO frame the active site entrance and form a wide and surface-exposed funnel. In AnVHPO, however, it is mainly the upper loop that engages in tunnel formation in front of the active site, correlating with an onset of chlorination activity. This loop is in the same region as the non-resolved loop in wt-AmVHPO that becomes ordered in the R425S mutant, hinting at the influence of this structure on halogen selectivity. Also, in the bacterial NapH1, two prominent loops in the respective N-terminal part of each subunit strongly interact at the monomers' interface. They are responsible for the dense packing around the active site. Only the chloroperoxidase CiVHPO acts as a monomer but likewise uses a loop structure in front of the active site that closes off the entrance and takes part in tunnel formation. All these observations show that the defined loop formation and its induction to form a tunnel structure to the active site is a general scheme in VHPOs that regulate substrate specificity and reactivity. This further underlines the importance and broad application of our successful engineering approach.

Furthermore, the structural rearrangement induced by the single point mutation led to a defined specificity pocket inside the newly formed tunnel, enabling TMB (**7**) access and binding at the intersection of two neighboring subunits. The released hypochlorous acid can now immediately react with the substrate, thus further contributing to the enhanced chlorination ability of R425S-AmVHPO. The structural and

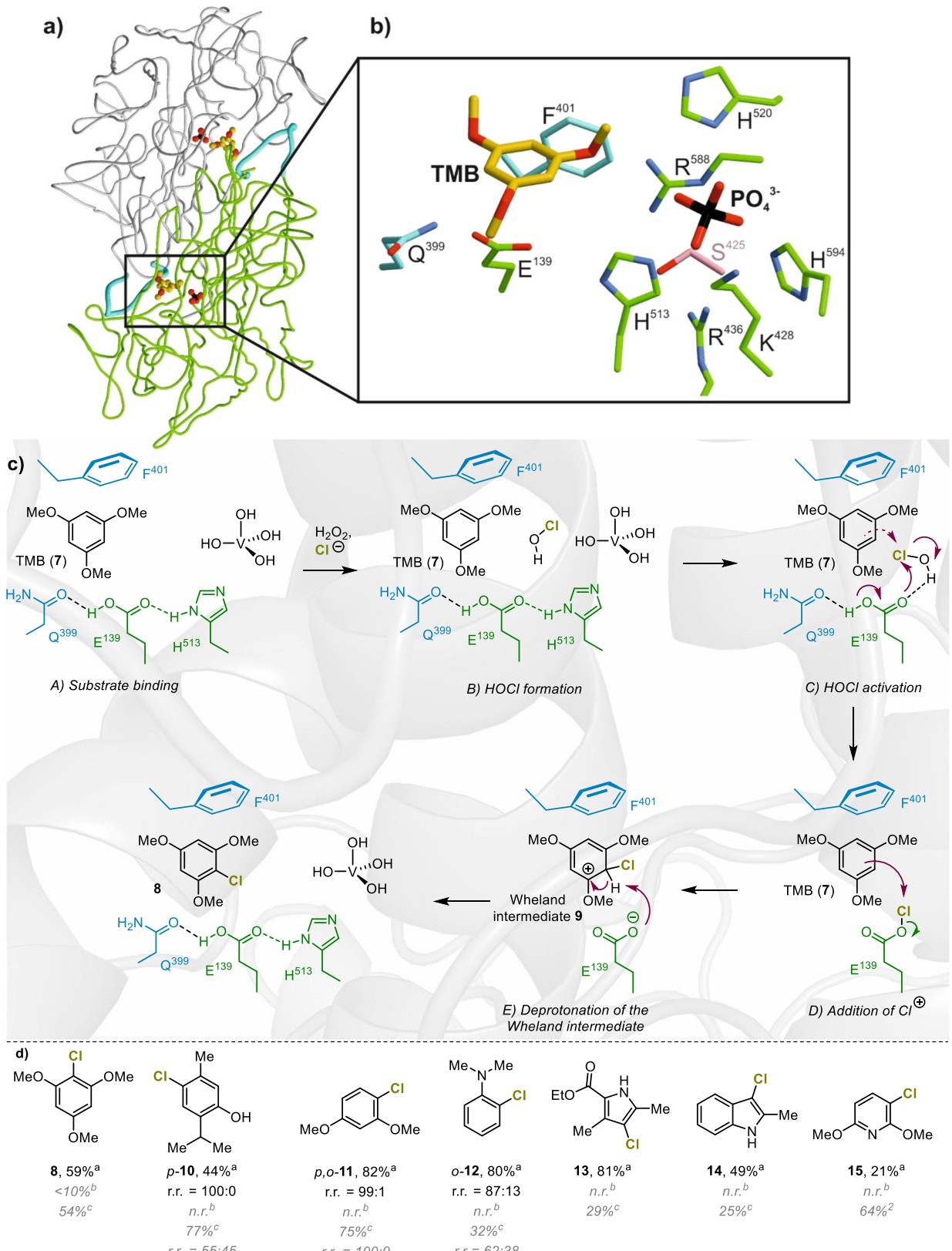

**Fig. 5 | The active site in *Am*VHPO is formed by two adjacent subunits. a** Coil representation of neighboring subunits in the *Am*VHPO:TMB-R425S complex. The phosphate mimicking the catalytic vanadate is coordinated by one subunit, while the substrate binding channel is formed together with the adjacent subunit. The loop region from the flanking subunit is crucial for TMB binding (loop region, residues 390–404, highlighted in cyan) and is structured only in the R425S mutant. **b** Close-up view of the active site with protein side chains engaged in ligand and phosphate binding. S425 is colored pink. **c** Proposed schematic mechanism of enzymatic chlorination in R425S-*Am*VHPO. **d** Preliminary substrate scope of the R425S-catalyzed chlorination in comparison with those addressable by wild-type *Am*VHPO and NCS/ DABCO. [a]isolated yield using R425S-*Am*VHPO as catalyst; [b]yields determined by LC-MS or GC-MS using wild-type *Am*VHPO as catalyst; [c]isolated yield using NCS/DABCO; n.r. no reaction; r.r. ratio of regioisomers. TMB = 1,3,5-trimethoxybenzene; NCS/ DABCO = N-hydroxysuccinimide/1,4-Diazabicyclo[2.2.2]octan.

**Vanadium-dependent Bromoperoxidases**

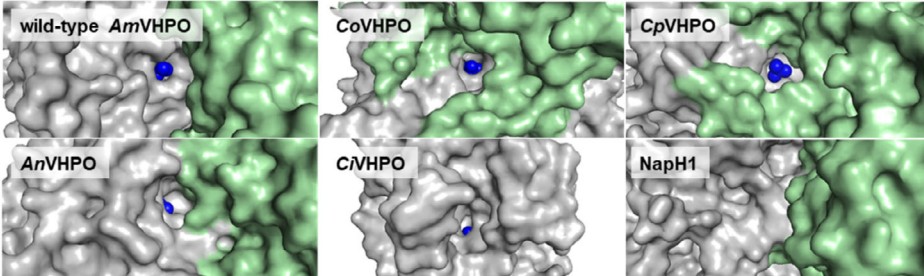

**Vanadium-dependent Chloroperoxidases**

**Fig. 6 | Exposure of the active site cofactor to the surface in different VHPOs.** The bromoperoxidases wild-type *Am*VHPO (PDB: 5LPC), *Co*VHPO (PDB: 1QHB), and *Cp*VHPO (PDB: 1UP8) and the chlorinating enzymes *An*VHPO (PDB: 1QI9), *Ci*VHPO (PDB: 1IDQ), NapH1 (PDB: 3W36) are compared. Enzymes are depicted as dimeric units (gray and light green), except *Ci*VHPO (PDB: 1IDQ), which mainly acts as a monomer. In the case of the dodecamers wild-type *Am*VHPO (PDB: 5LPC), *Co*VHPO (PDB: 1QHB), and *Cp*VHPO (PDB: 1UP8), a combination of two subunits was chosen that resembles the densest packing around the prosthetic vanadate and phosphate group, respectively (blue spheres). VHPO = vanadium-dependent haloperoxidase; *Am*VHPO = vanadium-dependent haloperoxidase from *Acarychloris marina*; *Co*V-HPO = vanadium-dependent haloperoxidase from *Corallina officinalis*; *Cp*VHPO = vanadium-dependent haloperoxidase from *Corallina pilulifera*; *An*VHPO = vanadium-dependent haloperoxidase from *Ascophyllum nodosum*; *Ci*VHPO = vanadium-dependent haloperoxidase from *Curvularia inaequalis*; NapH1 = vanadium-dependent napyradiomycin haloperoxidase from *Streptomyces sp. CNQ-525*.

biochemical studies complemented by MD simulations highlight Phe401 and Glu139 as crucial residues for substrate binding and halogenation activity, with a significant or even total activity loss upon their exchange by glycine. Remarkably, glutamic acid also plays a dominant role in FAD halogenases and is crucial for high reaction rates[55,56]. For example, the single point mutation E346Q in the flavine-dependent halogenase PrnA caused a reduction in halogenation rate by two orders of magnitude[57,58]. Theoretical investigations and active site mutagenesis studies showed that localization of negative charge and the interaction with the chlorinating species is essential for the electrophilic aromatic substitution in such enzymes. In our R425S variant, the hypochloric acid formed at the vanadate cofactor coordinates to Glu139 via hydrogen bonding, thus shuttling Cl+ from the vanadate binding site to the substrate binding site. Simultaneously, Glu139 increases the electrophilicity by this hydrogen-bonding interaction or even by forming the glutamyl hypochlorite intermediate (D, Fig. 5c). Both activation modes will lead to an electrophilic addition of the aromatic ring to the corresponding hypohalite species (dashed versus plain arrows, C & D, Fig. 5c). It is plausible that the E139G variant could not chlorinate TMB (7) due to lower electronic activation of the substrate compared to 3 and 5 (see SI, chapter 4). After the electrophilic Cl+ is transferred to TMB (7), Glu139 stabilizes the cationic Wheland intermediate (9, Fig. 5c) via ionic interactions, and the carboxylate moiety in Glu139 facilitates deprotonation of cationic intermediate furnishing the final product 8.

In summary, our results provide insights into the long-standing question of halogen specificity in VHPOs and have proven the existence of a substrate-binding site in VHPOs for the first time. An AI-guided rational design led to a successful enzyme engineering to switch the halogen specificity in *Am*VHPOs. A single point mutation of an amino acid residue outside the active site (position 425) in *Am*VHPO initiates a complex structural rearrangement within the protein scaffold, leading to an engineered enzyme pocket, which enables efficient aromatic chlorinations. Comparing the structures of different VHPOs points toward a general correlation between the chloroperoxidase activity and the enclosure of the active site prosthetic group within the enzyme, which is influenced by loops surrounding the entrance to the active site that is mainly located at the interface of two neighboring protein subunits. Halogen oxidation and halogen delivery in VHPOs are locally separated and occur at different binding sites that are closely related. While the vanadate binding site is responsible for halogen oxidation, another binding site in the surroundings accommodates only specific substrates. Larger substrates, which have been substrates

to the wild-type *Am*VHPO, are not converted by the mutant enzyme because of the now-defined space within the tunnel structure. In addition, a glutamic acid residue within the substrate binding site plays a decisive role in the chlorination ability of aromatic substrates. These structural features are similar to those identified in FAD-dependent halogenases and corroborate the evolution of a similar concept by nature for selective halogenations in both enzyme classes. Overall, the mechanistic insights presented in our study advance the general understanding of oxidative halogenations in VHPOs. This gives us a handle for easily classifying and manipulating wild-type VHPOs and tailoring them to the needs of modern organic chemistry. Given the advantages of VHPOs for organic synthesis compared to classical chemical catalysts, especially regarding sustainability and environmental protection, we are convinced that our findings will lay the foundation for their further engineering and, thus, biotechnological application in the future.

## Methods
### Cloning and site-directed mutagenesis
The codon-optimized synthetic gene of the haloperoxidase I from *Acaryochloris marina* (wild-type *Am*VHPO), cloned into a pET28-based expression vector, was used as a template for site-directed mutagenesis experiments.[1] All mutants of the *Am*VHPO were created by PCR using primers from STable 1. Genes for wild-type and mutants of *Co*VHPO and *Cp*VHPO were synthesized and cloned into pET28b(+) (Novagen, 69865-3) by Azenta and were confirmed *via* Sanger sequencing (GATC *Biotech*; Azenta).

### Activity screening of *Am*VHPO single-point mutants
A site-saturated mutagenesis library of position R425 was ordered from ThermoFisher Scientific. The mixture of all mutants was heat-transformed into freshly prepared chemo-competent *E. coli* (BL21) cells and spread afterward on kanamycin-containing lysogeny broth (LB, Agarose Sigma, A9539, LB Medium Sigma, L3022) agar plates. A 2 mL well-plate with 94 single-picked colonies was incubated overnight (37 °C; 900 rpm). For expression of the enzyme, a second plate was prepared, inoculated, and incubated at 37 °C for 90 min before the expression (18 h, 18 °C, 900 rpm) was induced with IPTG (Sigma-Aldrich, I6758, 0.1 mM). After 18 h, the cells were centrifuged (2000 × *g*; 10 min), resuspended in Tris (Fluka, T4661) pH 7 (50 mM Tris, 300 mM NaCl (Sigma, S9888), 10 mM imidazole (Carbolution, CC03002), 10% (v/v) glycerol (Alfa Aesar, A16205), and lyzed by the freeze-thaw method. The overnight cultures were also used to prepare

a glycerol stock master plate. To detect all mutants with increased activity, the cultures were centrifuged (2000 × *g*; 10 min) again, and a monochlorodimedone (**3**) (MCD, Carbolution, CC03081) assay was performed with the lysate. Therefore, 365 μL of a stock solution containing MCD (**3**, 50 μM), MES (TCI, M0606) pH 6 (50 mM), KCl (Sigma, P4504) or KBr (Acros, AC196480010) (200 mM), and Na$_3$VO$_4$ (Sigma, 450243) (300 μM) was mixed with 5 μL of the lysate. The measurement was started by adding 30 μL H$_2$O$_2$ (Acros, 202460010) (10 mM) to the reaction mixture, and a decrease in absorbance was monitored at 290 nm and 30 °C for 30 min. Active mutants were identified by Sanger sequencing.

## Expression and purification

Confirmed mutagenesis products were transformed into *E. coli* BL21 (DE3) using a standard heat shock protocol. Recombinant bacteria were grown on LB agar plates and amplified afterward in LB medium at 37 °C to an optical density of 0.6 in the presence of kanamycin (50 μg/mL). The protein expression was induced by IPTG (1 mM), and the culture was incubated at 18 °C overnight. Cells were collected by centrifugation, washed with saline (0.9% NaCl), and resuspended in buffer A (50 mM Tris pH 7.0, 300 mM NaCl, 30 mM imidazole). Before sonication of the harvested cells, DNase I (ThermoFisher, EN0521) and protease inhibitors (PMSF, ThermoFisher, 36978) were added. After centrifugation, the supernatant was heated to 60 °C for 20 min to precipitate unwanted proteins. Soluble target proteins, carrying a His-Tag sequence, were purified by an Äkta purifier using a 5 mL Histrap FF column (GE Healthcare) and an imidazole gradient from 30 to 500 mM. After analyzing the protein peak fractions by SDS-PAGE, haloperoxidase fractions were concentrated by ultrafiltration and dialyzed against buffer B (20 mM Tris pH 7.0, 150 mM NaCl) for a size exclusion chromatography (S200 16/60 column, GE Healthcare). Target proteins were dialyzed against buffer C (50 mM Tris pH 7, 100 μM Na$_3$VO$_4$) for storage and further use.

## Monochlorodimedone assay

All reactions were performed in triplicates using 96-well microplates (Brand® F-bottom, UV-transparent, pureGrade™, 781600). A reaction mixture containing MES buffer pH 6 (50 mM), KCl (200 mM), Na$_3$VO$_4$ (1 μM), and the respective *Am*VHPO variant (4.4 μg/mL) was prepared, and MCD (**3**, 50 μM; 1 mM stock solution in 2 M NaOAc (Fluka, 32319)) was added. The volume per well was 270 μL. To start the measurement, 30 μL H$_2$O$_2$ (10 mM, final concentration 1 mM) was added last, and the decrease in absorption (290 nm) was monitored over time (30 min) at 30 °C by UV/Vis spectroscopy. Data were analyzed using Microsoft® Excel® 2019 MSO version 16.0.10366.20016 and OriginPro 2019 version 9.6.0.172.

## Phenol red assay

All reactions were performed in triplicates using 96-well microplates (Brand® F-bottom, pureGrade™). Phenol red (**5**, Alfa Aesar, B21710) 14.3 μM; stock solution prepared in 100 mM NaOH) was added to a mixture of MES buffer pH 6 (50 mM), KX (142 mM, X = Cl, Br), Na$_3$VO$_4$ (300 μM) and the respective *Am*VHPO variant (for bromination 6.67 μg/mL; for chlorination 33.3 μg/mL). The reaction was started by the addition of H$_2$O$_2$ (10 mM), and the absorption (590 nm) was measured over time (bromination 10 min; chlorination 45 min) at 30 °C. Data were analyzed using Microsoft® Excel® 2019 MSO version 16.0.10366.20016 and OriginPro 2019 version 9.6.0.172.

## 1,3,5-Trimethoxybenzene (7, TMB) conversion monitored by LC analysis

High-performance liquid chromatography was performed on an Agilent 1260 Infinity II system. A reaction mixture containing MES buffer pH 6 (55 mM), KCl (550 mM), Na$_3$VO$_4$ (1.1 μM), 200 μM TMB (Sigma, 138827), and the respective *Am*VHPO variant (13.2 μg/mL)

was prepared and incubated in a shaker at 30 °C and 850 rpm. 110 μL H$_2$O$_2$ (10 mM, final concentration 1 mM) was added to start the reaction. 100 μL samples were taken at indicated time points. Reactions were quenched by adding 25 μL of sat. NaCl and 25 μL of sat. (NH$_4$)$_2$SO$_4$ (ThermoScientific, 10556792). 1.5 μL of a 3 mM phenol (Sigma, W3222318) stock solution (internal standard) was added, and samples were centrifuged and subjected to LC-MS analysis. Samples were analyzed using 2–70% or 80% acetonitrile (ThermoScientific, 10001334) gradient in 0.1% aqueous formic acid (Sigma, 5.330002) over 7 min. Chromatograms were extracted at 266 nm. Peak areas (AUC) were determined and referenced to the internal standard (phenol). t$_R$ (phenol) = 2.7 min, t$_R$ (TMB **7**) = 5.4 min, t$_R$ (chlorinated TMB **8**) = 5.7 min. Assays were performed as triplicates. Data were aquired using OpenLab CDS Acquisition version 2.6 and OpenLab CDS Analysis version 2.6. Data were analyzed using Microsoft® Excel® 2019 MSO version 16.0.10366.20016 and OriginPro 2019 version 9.6.0.172.

## Enzyme kinetics

A 1 mM stock solution of MCD was prepared in 2 M NaOAc. The 270 μL reaction mixture consisted of MES buffer (50 mM, pH 6.0), MCD (50 μM), KBr (200 mM) or KCl (200 mM), Na$_3$VO$_4$ (1 μM), and R425S-*Am*VHPO (final concentration 4.4 μg/mL). For Michaelis-Menten kinetic studies for H$_2$O$_2$ consumption, H$_2$O$_2$ concentrations were varied in a 2-fold dilution series starting from 10 mM to 0.039 mM (the concentration of the 100 mM stock solution was verified by titration using KMnO$_4$), while keeping KCl concentration at 2 M. Final enzyme concentration for R425S-*Am*VHPO was 4.4 μg/mL. For Michaelis-Menten kinetic chlorination studies, KCl concentrations varied from 50 to 2000 mM while keeping the H$_2$O$_2$ concentration at 1 mM. The final enzyme concentration for R425S-*Am*VHPO was 4.4 μg/mL. For Michaelis-Menten kinetic studies for bromination, KBr concentrations varied from 6.25–1200 μM, keeping the H$_2$O$_2$ concentration at 1 mM. The final enzyme concentration for R425S-*Am*VHPO was 2 μg/mL. The decrease in absorbance at 290 nm was monitored continually while keeping the temperature at 30 °C and mixing at 410 rpm. Pseudo-first-order kinetics were applied. All assays were performed in triplicates in 96-well polystyrene microplates (Brand® F-bottom, UV-transparent, pureGrade™). The concentration of converted monochlorodimedone **3** was calculated from A$_{290}$ using the following equation (Eq. 1).

$$c(MCD) = \frac{A_{290}}{\varepsilon_{MCD} \cdot l}$$

$$\varepsilon_{MCD} = 19.9\,\text{mM}^{-1}\text{cm}^{-1};\ l = 0.91\,\text{cm} \tag{1}$$

Data were analyzed using Microsoft® Excel® 2019 MSO version 16.0.10366.20016 and OriginPro 2019 version 9.6.0.172.

## Competition assay for chlorination of MCD (3) and TMB (7)

A 2 mM MCD (**3**) stock solution in 2 M NaOAc and a 2 mM TMB (**7**) stock solution in DMSO (Acros, 390750010) was prepared. Into 875 μL of a master mix containing MES buffer (50 mM, pH 6.0), KBr (200 mM, for wt-*Am*VHPO) or KCl (200 mM, for R425S-*Am*VHPO), Na$_3$VO$_4$ (1 μM), and the respective enzyme (final concentration 4.4 μg mL$^{-1}$), 100 μL of a TMB (**7**) solution in DMSO was added and incubated for 30 min at rt. Initial concentrations of TMB (**7**) solutions in DMSO were 2 mM, 1 mM, 0.75 mM, 0.5 mM, 0.25 mM, 0.125 mM, and 0 mM (pure DMSO). After incubation, 25 μL of the MCD (**3**) stock solution was added. 270 μL of this solution (triplicates) were pipetted into a 96-well polystyrene microplate (Brand® F-bottom, UV-transparent, pureGrade™), and the reaction was started by the addition of 30 μL 10 mM H$_2$O$_2$. The decrease in absorbance at 290 nm was monitored continually while keeping the temperature at 30 °C and mixing at 410 rpm. All assays were performed at least three times in triplicates. Pseudo-first-order kinetics were applied. Initial velocities of MCD consumption in the

presence of TMB (**7**) were plotted against the ratio of TMB (**7**) to the sum of MCD (**3**) and TMB (**7**). Data were analyzed using Microsoft® Excel® 2019 MSO version 16.0.10366.20016 and OriginPro 2019 version 9.6.0.172.

## Computational screening for mutation sites close to the vanadate binding site

**Structure preparation.** Due to the lack of resolution of the loop at positions 389 to 405 in the PDB, 5LPC crystal structure of the bromoperoxidase[21] AlphaFold2[59] was used to create a model of the monomer structure. Complex assembly was performed with the Rosetta suite of biomolecular modeling software[60] (v.3.13 2021.16+release). A symmetry definition file was created[61] based on the biological assembly of PDB: 5LPC and used to construct a dodecamer consisting of twelve copies of the AlphaFold2 model. The model was minimized with symmetry definitions to resolve potential clashes in the structure.

**Sequence design.** The sequence was designed step by step with the deep learning-based method ProteinMPNN[41]. First, the designable space was reduced. For this, the distance to the catalytic residue K428 (max 10 Å) as well as the solvent-accessible surface area (SASA > 0) were used as filters to select positions for the design. Residue distances and SASA were calculated with the Biopython library[62], the latter using the ShrakeRupley[63] class. Sequence design was performed position by position, keeping all other residue identities fixed. Conditional probabilities were predicted for the predicted monomer and composite dodecamer structure and the 5LPC crystal structure with a temperature of 0.3.

**ΔΔG calculations.** Structure optimization and energy calculations were performed in the Rosetta suite. ΔΔGs were computed following the Rosetta implemented cartesian ΔΔG protocol[64]. The dodecamer structure was relaxed in cartesian space, and the ΔΔG for every amino acid substitution was calculated for the 5 best scoring structures out of 500. For each structure and mutation, 10 ΔΔG iterations were performed. The best-scoring mutant out of 10 was used for energy and SASA calculations.

**Visualization.** Structural visualizations were created with chimerax[65].

**Molecular dynamics (MD) simulations.** To explore the conformational landscape of the loop region spanning position 389 to 405 in the wild type compared to the R425S variant, Hamiltonian replica exchange (HREX) MD simulations were conducted. The starting configurations of wild-type and variant were based on the R425S crystal structure. For the wild-type, the S425 residue was reverse-mutated to Arginine. Parameters for the Histidine-Vanadate complex were adapted from the literature[25]. HREX MD simulations were performed based on the protocol described in Schwarten et al.[66]. Simulation setup and production runs were carried out with GROMACS 2023.1[67]. Both the wild type and the R425S variant systems were modeled as homodimers, including the active center and substrate tunnel formed by the two monomers. Each system was solvated in a square water box with a side length of 127.3 Å. Na+ and Cl⁻ ions were added to neutralize the system at a salt concentration of ~200 μM. The AMBER99SB-ILDN force field[68] was used for molecular interactions, and the TIP3P[69] water model for solvation. In total, the system contained 62257 water molecules, 232 Na⁺ and 190 Cl⁻ ions in addition to the protein chains.

Minimization was carried out using a steepest descent protocol. Subsequently, two equilibration steps were performed: an equilibration under constant volume (NVT) and a second under constant pressure (NPT) at 300 K, with durations of 1 ns each, employing the v-rescale thermostat[70] and c-rescale barostat[71]. The same thermostat and barostat were employed during all production runs. Eight HREX

replicas were carried out, with temperatures spanning from 300 K to 457 K. The "hot" region included all atoms of residues Phe389 to Gly405. Each replica was simulated for 200 ns at constant pressure, with an average exchange rate of 0.36 to 0.41. This resulted in a cumulative trajectory time of 1.6 μs per system.

Principle component analysis was applied to pairwise Cα-distance matrices to characterize the diversity of the conformational landscape across the two simulations Loop conformations were clustered using the DBScan algorithm implemented in the scikit-learn package[72].

Setup and simulation data are available at https://github.com/ClaraTSchoeder/schoederlab.

## Crystallization

**Crystallization of *Am*VHPO variants.** *Am*VHPO (15 mg/mL) was crystallized by the hanging drop vapor diffusion method at 20 °C. For co-crystallization experiments, the surrogate 1,3,5-trimethoxybenzene (TMB, **7**, 100 mM stock solution in acetonitrile) was added to *Am*VHPO variants in a final concentration of 1 mM. Crystallization drops had a maximum volume of 2 μl with a 2:1 ratio of protein and reservoir solution (0.1 M Tris pH 8.5, 2 M NH₄H₂PO₄ (Sigma, 2004005), 0.5 mM K₃VO₄, FisherScientific, 11328308). Crystals were cryoprotected by a 7:3 mixture of mother liquor and 100% (v/v) ethylene glycol (Sigma, 293237) and subsequently vitrified in liquid nitrogen.

**Structure determination of *Am*VHPO variants.** Datasets of *Am*VHPO variants were recorded with synchrotron radiation of $\lambda = 1.0$ Å at the beamline X06SA, Swiss Light Source (SLS), Paul Scherrer Institute, Villigen, Switzerland. Reflection intensities were evaluated with the program package XDS, and data reductions were carried out with XSCALE[73] (Table S5). Each structure was solved by Patterson search calculations with REFMAC5[74] and coordinates of *Am*VHPO (PDB ID 5LPC)[21]. The models were completed using COOT[75] in combination with REFMAC5, respectively. Notably, the *Am*VHPO_R425S was determined in the absence and presence of 1,3,5-trimethoxybenzene (TMB, **7**). Restrained and TLS (Translation/Libration/Screw) REFMAC refinements yielded excellent $R_{\text{work}}$ and $R_{\text{free}}$ values as well as root-mean-square deviation (rmsd) values of bond lengths and angles. All crystal structures have been deposited in the RCSB Protein Data Bank (Table S5).

## Aromatic chlorination catalyzed by R425S-*Am*VHPO

**General procedure.** R425S-*Am*VHPO variant (273.7 μg/mL) was pre-incubated with a 1.74 mM solution of Na₃VO₄. Substrate (7.43 mM; 11.9 μmol) was solved in an MES (50 mM pH 6) buffered solution of water and acetonitrile (1/1 v/v), containing KCl (4.4 eq., 32.5 mM) H₂O₂ (16.4 mM) and the enzyme in a total volume of 1.60 mL. The reaction mixture was incubated for 24 h at 30 °C. Afterward, halfmolar amounts of standard with respect to initial substrate concentration were added, and the reaction was stopped by adding a sat. NaCl/(NH₄)₂SO₄ solution and extracted with ethyl acetate. Standards were phenol (LC-MS), n-dodecanol (Sigma, 443816) (GC-MS) or 1,4-dibromobenzene (abcr, AB111795) (NMR). The solvent was removed under reduced pressure, and the product was analyzed without further purification. LC-MS data were obtained using OpenLab CDS Acquisition version 2.6 and OpenLab CDS Analysis version 2.6. GC-MS data were obtained using MassHunter Workstation GC/MS Data Acquisition version 10.2.489 and MassHunter Workstation Qualitative Analysis version 10.0. ¹H-NMR spectra are shown for the purified main products. NMR data were analyzed using MestReNova version 14.1.0-24037.

## Chemical chlorination

**General method.** A round bottom flask equipped with a stirring bar was charged with the arene/heterocycle (0.22 mmol, 1.0 eq.), N-chlorosuccinimide (NCS, Alfa Aesar, A10310, 1.1 eq.), and DABCO (Sigma-Aldrich, D27802, 5–10 mol%). Dichloromethane (Acros,

8812283) (5 ml) was added. The reaction vessel was sealed and stirred at ambient temperature overnight. The crude product was purified by column chromatography using silica gel with ethyl acetate in hexane as the eluent. LC-MS data were obtained using OpenLab CDS Acquisition version 2.6 and OpenLab CDS Analysis version 2.6. GC-MS data were obtained using MassHunter Workstation GC/MS Data Acquisition version 10.2.489 and MassHunter Workstation Qualitative Analysis version 10.0. $^1$H-NMR spectra are shown for the purified main products. NMR data were analyzed using MestReNova version 14.1.0-24037.

Further information on research design and methods is available in the supplementary data linked to this article.

### Reporting summary
Further information on research design is available in the Nature Portfolio Reporting Summary linked to this article.

## Data availability
The data generated in this study are provided in the Supplementary Information/Source Data file and can be obtained from the corresponding author on request. MD simulation data is publicly available at https://doi.org/10.5281/zenodo.14544279. Protein X-ray crystal structures have been deposited at the Protein Data Bank under the following accession codes: 5LPC (wild-type *Am*VHPO), 8Q2O (*Am*VHPO R425D mutant), 8Q21 (*Am*VHPO R425S mutant), 8Q22 (*Am*VHPO R425S mutant with TMB). Source data are provided with this paper.

## Code availability
The Rosetta software suite used for structure preparation and ddG calculations is publicly available under the RosettaCommons license. Specific scripts for the computational pipeline in this publication can be accessed at https://github.com/ClaraTSchoeder/schoederlab and are given in Supplementary Data.

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

## Acknowledgements

We thank Prof. T.A.M. Gulder for the fruitful discussions and Dr. A. Zlobin for his assistance with the simulation setup. This work was funded by the Emmy-Noether and Heisenberg program of the German Research Foundation (DFG, GU 1134/3-1 and GU 1134/4, T.G.). C.J.S. thanks the Deutsche Bundesstiftung Umwelt for a fellowship (grant 20015/400). T.G. thanks the PharmaScienceHub (PSH) for their support.

## Author contributions

P.Z. and K.B.S. planned and performed the biological experiments, analyzed the data, wrote supplementary information, prepared figures, and contributed. R.Z. performed protein expressions, biological assays, and chemical chlorination. C.J.S. performed the sequence alignment of

different VHPOs and initial experiments. V.M. and C.T.S. performed the computational optimization of the mutation sites and the MD simulations. M.G. conducted all work proteins with and without substrates. T.G. conceived and supervised this project, analyzed the data, and wrote the main manuscript. All authors contributed to the writing of the manuscript and the Supporting Information.

## Funding

## Competing interests
The authors declare no competing interests.
