## [Transparent Peer Review file · Nature Communications]

Unraveling the Molecular Basis of Substrate Specificity and Halogen Activation in Vanadium-Dependent Haloperoxidases

Corresponding Author: Professor Tanja Gulder

A version of this paper was originally rejected for publication by Nature Communications, however that decision was reconsidered after appeal by the authors.

Version 0:

Reviewer comments:

Reviewer #1

(Remarks to the Author)

Proposal Summary: Gulder and colleagues have presented an investigation that aims to gain mechanistic and structural insight into the halide oxidation and substrate binding requirements for catalysis in the vanadium haloperoxidase (VHPO) class of halogenase enzymes. Using the vanadium-dependent haloperoxidase from the cyanobacterium *Acarychloris marina* MBIC 11017 (AmVHPO) as their model enzyme, the authors used computational screening to identify mutation sites that could be modified without harming expression and are close to enzyme active site. Mutagenesis of an identified R425 residue that is i-3 from the active site demonstrated significant changes in both bromination and chlorination rates using a UV monochlorodimidone (MCD) assay. An R425S variant increased these activities from 53% yield and 8% yield to 91% and 74% yield, respectively. The authors demonstrated the R425S is also an efficient catalyst for the chlorination of other aromatic substrates. The authors crystallized the R425S mutant (highly reactive VCPO) and the R425D mutant and found that the R425S mutant showed an interaction between Ser425 and a region of the adjacent subunit (residues 390-404) creating a critical loop structure, supporting its vital role in halogen specificity for chloride. To gain insight into organic substrate binding, competition assays for chlorination were performed between 1,3,5-trimethoxybenzene (TMB) and MCD for R425S-AmVHPO showing a preference for TMB and suggesting substrate binding. The authors confirm binding of TMB to the R425S variant through x-ray crystallography. The residues that surround TMB were confirmed to be Glu139 and Gln399 and Phe401 from the neighboring subunit. Phe401 is proposed to anchor the substrate through a pi-pi interaction and Glu139 and Gln399 are thought to facilitate the electrophilic aromatic substitution process. The R425S E139G and R425S F401G variants were also prepared, and chlorination no longer took place with the F401G mutant, but only slowed down bromination, suggesting that binding the substrate is important for the chlorination process but less important for bromination. Decrease in chlorination activity and the lack of bromination with the E139G variant suggests the importance of a carboxylic acid moiety in hypohalous delivery to the substrate and possible contribution in stabilizing the Whealand intermediate prior to the deprotonation/aromatization process of the electrophilic aromatic substitution process.

Reviewer Commentary: Gulder and colleagues have presented an intriguing mechanistic and structural investigation into the required halide oxidation and substrate binding requirements for vanadium haloperoxidase enzymes. I would begin by acknowledging that this type of investigation is non-trivial and several of the mechanistic components (halide oxidation selectivity, halide binding, hydrogen peroxide binding, organic substrate binding, halogen occurring inside or outside of the active site) have been topics of discussion for a long time. The work presented here is remarkably exciting for numerous reasons including 1) the clever computational approach to design mutants of the VHPO that could play a role in gaining insight into the mechanistic nuances of catalysis, 2) the identification of the R425 position in playing a role in creating a structural loop that both allows for chlorination (vs bromination) and facilitating substrate binding by increasing the residence lifetime of a substrate, and 3) establishing the first crystal structure of a VHPO with an organic substrate bound. These components alone would warrant publication in Nature Communications.

Despite my excitement for this work, my only hesitation is the connection these findings make with wild type VHPO enzymes catalyzing these processes in nature. While I certainly agree that the R425 mutation is very important to have identified and provides insight into the enzyme having a sophisticated way of creating an organic substrate binding site, I think it's critical that the authors spend some more time framing what these results mean (or could mean) for wild type VHPO enzymes prior

this work being accepted for publication. I congratulate the authors on the wonderful insights provided and I do recommend publication in Nature Communications after the following items have been addressed:

Main concerns to address:

- Was x-ray crystallography attempted with the wild type enzyme with the TMB substrate? What confirmation is there that TMB doesn't bind? It would be helpful to know that this was attempted, and the crystal structure did not have TMB incorporated, for example.
- Have any of these structures been collected with bromide bound AND organic substrate bound? I ask this based on the proposal by Littlechild (Journal of Inorganic Biochemistry, 2009, 103, 617-621.) where it is suggested that bromide binding leads to the formation of a hydrophobic tunnel that facilitates organic substrate binding.
- If it's been established that the wild type AmVHPO can brominate TMB (ChemBiochem, 17, 2028-2032), then I'm trying to understand how the authors think this is occurring. Presumably it is through a binding event followed by bromination, but how would it do this if the TMB doesn't have a long enough residence time? I think the authors do allude to some "on the surface" proposals, but having a more concrete section on this would be informative for the reader.

Some additional items:

- I'd like to request that the authors include control runs (no enzyme, no vanadate salt, no halide, no hydrogen peroxide) for their investigations in the supporting information.
- Line 152 in the manuscript has hydrogen peroxide listed after "(see SI, chapter 5)". Was this intentional?
- Despite it not being the focus of this study, I would suggest adding a figure with the other substrates tested in the core manuscript rather than the SI. (discussed in lines 177-184 in the manuscript). As part of this, the addition of what the results look like with the wild type enzyme would be interesting for direct comparison.
- I was curious if the authors looked for any regioselectivity trends with the R425 mutants on 1,3-dimethoxybenzene to see if it plays any role in modulating regioselectivity.
- Despite the encouraging results with the TMB vs MCD competition experiments, is there literature that could be referenced that looks at the relative halogenation rates of TMB (or other substrates that have been used in the same type of experiment) with a standard halogenating agent like N-bromo/chlorosuccinimide? My hunch is that all these substrates might just be better inherent nucleophiles than MCD and that these competition experiments aren't as suggestive as the community anticipates they are. In this case, I would suggest either finding that data or just running the comparative halogenation reaction between TMB and MCD with NBS/NCS to ensure the trend you're seeing isn't just differences in nucleophilicity of the substrates. I should mention that I liked this competition experiment, but this would be just to provide a little more insight into how fair this competition is.

Reviewer #2

(Remarks to the Author)

Gulder et al introduce single point mutations to the cyanobacterium *Acaryochloris marina* vanadium dependent bromoperoxidase AmVHPO at position R425. This residue was identified using models produced by ProteinMPNN and subjected to site saturation mutagenesis. Many of the mutations led to an increase in both bromination and chlorination activity. Five aromatic substrates could be chlorinated with a level of regioselectivity, conversions of 24 – 95%, and basic kinetics were performed. Crystal structures of two mutants, R425S and R425D, were solved. R425S mutation resolved a previously flexible loop 390-404 between enzyme subunits which allowed an aromatic substrate to bind, and somewhat reduced solvent accessibility to the active site. Residues close to the "binding site" E139 and F401 were also mutated, leading to a general decrease in halogenation activity.

While the identification of a second sphere residue mutation able to dramatically increase the level of chlorination is useful, the claims that this mutation, and the more resolved loop or "binding pocket" that it creates is a basis for molecular chlorination in V-HPOs seems to be overreaching. The authors suggest that solvent accessibility to the vanadate binding site is reduced upon this loop ordering giving more time for hypochlorous acid to react with the substrate, and claim that other chlorinating V-HPOs contain a similar site which blocks the active site. No evidence, other than visual inspection of enzyme surface structure, is given to support this and thus further supporting evidence with additional chlorinating V-HPOs would be required to warrant publication in Nature communications.

Further specific comments included below:

Main Text

Please use either the three letter amino acid code or one letter code consistently throughout the manuscript.

Line 44 – 46 "Although recent achievements in homogeneous catalyst development have led to more selective catalytic halogenation approaches, these catalysts tend to be intricate, expensive, and time-consuming to synthesize." – references should be provided

Line 48 – 50 "The most common strategy in nature is electrophilic halogenation facilitated by vanadium- (VHPOs) and heme-dependent haloperoxidases, and flavin-dependent halogenases." – references should be provided and it should be mentioned that heme-dependent haloperoxidases have a similar mechanism to VHPOs

Line 95-97 "Mutagenesis experiments in the second shell combined with activity screening assays revealed that R425S-VHPO was transformed into a chloroperoxidase capable of efficient aromatic chlorination, even on a preparative scale." This suggests that there was no chlorination activity for the wild type enzyme. No mention of isolated yield for preparative scale experiments.

Lines 122, 161, 292 Naming position 425 as "i-3" is confusing and unnecessary. It also sometimes becomes "i+3".

Line 141 – 145 Data confusing and contradictory. SI figure 3 shows no activity for the chlorination of monochlorination with

wild type, here authors state 8% chlorination of wild type. Errors should be included in figure 3 data.

Line 163-165 "As different amino acids three positions apart from the active site lysine (Lys428) trigger an enhanced chlorination activity in different VHPOs, such as, e.g., Phe and Trp in CpVHPO, Trp in AnVHPO or Ser in AmVHPO"-References need to be provided, has the "enhanced chlorination" be probed previously?

"Fig. 3. A) The serine variant stands out with a ten times higher conversion of MCD (3) than the wild-type." - Needs to be made clear that this is only for chlorination, not bromination.

Fig. 3. D) – Not convincing data, why does conversion decrease between time points? Error bars need to be made much clearer

Line 181 – 184 "Intriguingly, a range of other aromatic substrates, such as TMB (7), thymol indoles, pyridines. were transformed to the corresponding chlorinated products with excellent regioselectivities and high yields (see SI, chapter 10)." - Actually a very small range of substrates, only 5, with ranges of conversion between 24 - 95%, 2 show regioisomers formed. Would not call this excellent regioselectivity or high yields. Indeed no isolated yields are reported.

Line 221 – 223 "However, for the highly reactive hypochlorous species HOCl, the chlorination reaction likely occurs immediately after the in-situ formation of the electrophilic chlorine reagent, rendering the halogenation of an enzyme-bound substrate very likely"- based on what? Reference for the comparative reactivity of HOCl and HOBr?

Line 225 – 226 "The hypothesis of a substrate binding site in R425S-AmVHPOs was thus further verified." – Overreaching statement

Fig. 5. - This figure is not effective at showing that TMB only binds to the R425S mutant. Why are superpositions of the liganded R425S with wt and R425D included? Currently it just looks like the substrate is bound in every enzyme. There should be a clear figure showing that only R425S binds the substrate in this way. If R425S binds the substrate, what is the binding constant?

Line 265 -268 - "A significant decrease in chlorination activity was indeed observed in the MCD (3, conversion of 3 was decreased by 20% after 15 min) and in the phenol red assay. At the same time, no bromination was detectable at all. Moreover, the chlorination of TMB (7) was only detectable in traces according to GC analysis" – This is confusing and needs to be simplified, essentially E139G reduces activity for both chlorination and bromination?

Line 270 Wheland spelling is incorrect

Line 302 – 305 "the Ser425 from the adjacent subunit alters the hydrogen-bonding network within the active site, thus forming a tunnel protecting the vanadate in the active site. This reduces the access of solvent molecules to the vanadate cofactor and thus abolishes the degradation of the reactive Cl⁺ species"- This appears to be speculative and further evidence should be provided, minimally a model

Supplementary Information

Chapter 6- Competition assays, S11, Showing that as increasing amounts of TMB is added to the reaction the consumption of MCD is slowed- how does this show that there is any TMB binding rather than just showing the equivalents of HOCl are getting used up by added TMB?

Chapter 9- Comparison of structure, It would be good to include some of these figures in the main text, as they are the main argument for active site accessibility controlling chlorination activity.

Chapter 10- Aromatic Chlorination, Protocol does not make clear if halogenated products have been extracted/purified, what the isolated yields were, or what the NMR analysis is of. Presumably the crude extract? Copies of the NMR trace should be included and references to literature data. There appears to multiple compound numbers for the same compounds? Many LCMS traces have multiple uncharacterised peaks. Are these side products? What are they? All data should be presented in standard formats.

Reviewer #3

(Remarks to the Author)

Manuscript "Unraveling the Molecular Basis of Substrate Specificity and Halogen Activation in Vanadium-Dependent Haloperoxidases" is a study of a vanadium-dependent haloperoxidase, where the authors attempt to explain halide preference of this class of halogenase, by biochemical and complementary structural studies. This work is well executed at the technical level. The authors mutate an arginine in the general vicinity of the active site, which makes the enzyme more permissive towards aromatic chlorination, in addition to maintaining robust bromination. The structural studies of the mutants show that the loop containing the Arg becomes ordered in that mutant. In my view, the level of novelty in this study is somewhat dampened by a prior study (cited here) where mutations of the analogous Arg in a homologous enzyme increased chlorination efficiency. Furthermore, the observation of the loop ordering per se is not sufficient to explain the halide reactivity without other studies, such as MD simulations. This is especially important, because the ordered loop contacts another subunit of the enzyme, so the ordering itself could be a result of the proximity to that subunit, and not directly related to the enzymatic activity changes. It is true that the ordering effect on activity is in any case allosteric, but how? The answer to this question backed up by other studies would be very insightful. Short of having this answer, the work is still largely observational, even if it brings in an important structural aspect. The writing style and illustrations somewhat lack clarity, as explained below for some instances.

Overall, this is a solid study suitable for a strong biochemistry journal, but, in my opinion, it does not rise to the level of Nature Communication.

Other comments:

Fig. 1: In panel a) the oxygens and the hydrogens need to be balanced properly on the right- and the left-hand sides, so please include additional product species. Somewhat related to this issue, this chemical scheme needs to be redrawn in a

sequential way. The formation of HOX is in the second reaction step, not in the reverse step of the first reaction (if the reaction was reversible as drawn, H₂O₂ would be the product of the reverse step).

Fig. 1 needs to be conventionally formatted, without panel headings and embedded questions.

Panel d) looks like an online abstract, rather than a figure panel. This figure additionally needs a major revision: the structure and its elements are not well depicted, and they may need to be shown as a cartoon with zoomed in regions as insets (instead of the text there), with judicious use of color. This is also an issue for Fig. 4.

The color labels will likely need to be removed from the figure legend and described in words.

Lines 91-108 are largely a repeat of the Abstract. Normally sentence in line 90 should conclude the Introduction.

Line 113-114 and then again in line 119. Need a reference to structures. Because you used the experimental structure as a starting point, you need to state so in line 119.

Line 120: what is meant by "predicted to respond well". What is the response being observed?
Similarly, in line 129, what is meant by "energetic advantage"? Is this borne out experimentally?

Line 143: you are referring to relative conversion- this is not the reaction rate.

Minor comments:

Line 67: reword "low halogenation reactivity of HOX in solution" to "low halogenation activity and chemical instability of HOX in bulk solvent";

Line 69, Fig. 1 legend and elsewhere: use "superimposition" instead of "overlay" or "alignment", when describing structure;

Line 76: please use a concrete scientific term instead of "oxidizing power", i.e. catalytic efficiency, catalytic activity, substrate scope, etc, as applicable here.

Line 77: the term "outer sphere" needs to be explained structurally. Or is this term just used to mean residues that are not conserved, but are in the vicinity of the catalytic residues? Please clarify.

Line 99: did the authors mean "intrinsically disordered"?

Fig. 3: absorbance, not absorption.

Version 1:

Reviewer comments:

Reviewer #1

(Remarks to the Author)

Gulder and coworkers have provided a substantially improved manuscript outlining their efforts to gain mechanistic and structural insight into halide oxidation selectivity and organic substrate binding requirements for catalysis by vanadium-dependent haloperoxidases (VHPOs). Building on their findings of a R425S variant of the vanadium-dependent haloperoxidase from the cyanobacterium *Acarychloris marina* MBIC 11017 (AmVHPO) that introduced a critical loop structure that is important from substrate binding and increasing bromination and chlorination activity, they have since shown that this same mutational effect is general across two other enzymes [Corallina pilulifera (CpVHPO-R397S) and Corallina officinalis (CoVHPO-R396S)]. From a VHPO tunability standpoint, this result on its own is a remarkably important finding that will play a significant role in future VHPO work going forward for this team, among others. I also found it very interesting during this revision that the R425S variant of AmVBPO for N,N-dimethyl aniline and thymol. To me this provides indisputable evidence for a tunable organic binding site that will be very exciting for the biocatalysis/organic synthesis community!

While I am a little concerned about the inability to co-crystallize AmVHPO and its mutants with bromide and organic substrate bound, it seems to me that there are likely some more dynamics that need to be uncovered in these systems, as Littlechild et al observed something very similar in their studies in 2009 (Journal of Inorganic Biochemistry, 2009, 103, 617- 621.).

I am remarkably impressed by Gulder and team's efforts to address our comments/suggestions (and others) and would now strongly recommend this report for publication in Nature Communications. Congratulations to Gulder and her team on this accomplishment!

Reviewer #2

(Remarks to the Author)

Unraveling the Molecular Basis of Substrate Specificity and Halogen Activation in Vanadium-Dependent Haloperoxidases.

The authors have addressed some of the points raised in the earlier review and additional data has been added to support their hypothesis and core claim regarding the existence of an (aryl-) substrate binding site in VHPOs. The authors did however fail to address many of the points this reviewer raised.

A structure and mutagenesis study, that produced the mutant R425S that led to more defined structure of a loop between residues 390 - 404, provided preliminary evidence for substrate accommodation within the enzyme's active site. The supporting MD simulations, the added results of transferring the R425S mutation to two VHPO homologs, as well as the additional mutagenesis studies of other residues in the surrounding region of the protein are promising steps toward demonstrating the mutation's general role in enhancing chlorination activity inside a binding site.

However, the manuscript, in its current form, still lacks definitive experimental evidence to support the claim that substrate binding occurs within the active site during the chlorination step. Unfortunately, the co-crystallization of the substrate 7 with the protein was not fruitful. Thus, the key experimental data currently supporting this hypothesis are the results from the competition assay between substrates 3 and 7 (SI, Chapter 6). Unfortunately, as currently presented, these assay results are difficult to interpret and do not yet convincingly demonstrate that the enzyme accommodates the substrate during the chlorination step. In addition, the authors provide no other explanations or experimental data in response to our comments on this matter.

If the manuscript were to be considered for publication, the authors would need to improve the clarity of the kinetic data on substrates 3 and 7. Specifically, it would be beneficial to establish whether these substrates act as competitive inhibitors to each other, rates of production of products, and kinetic data on binding efficiencies, which would provide more direct evidence of active site accommodation. This effect should be readily capturable via the initial velocity measurements. Additionally, if available, the authors might consider using a complementary technique, such as circular dichroism spectroscopy or a thermal shift assay, (or other) to provide evidence of binding. Alternatively, the authors may wish to reconsider their claims. In any event I am not sure the manuscript would meet the threshold for publication in Nature communications.

Reviewer #3

(Remarks to the Author)

The authors have adequately addressed my major comments.

Reviewer #4

(Remarks to the Author)

As a new reviewer, I was asked to only evaluate the new MD simulation data and resulting conclusions.

The authors claim that their MD simulations of the wt and R425S variant show "distinctly different conformational space", and I note that their analysis is limited to the 389-401 loop.

The supporting data (Figure S19, which is mis-cited in the main text as Figure SX) do show that the R425S loop sampling less conformation space than the wt loop, and having a lower RMSF across most of the loop residues.

However, the wt simulation is built on a model of the 425S variant, so without evidence that the wt simulation is adequately equilibrated, it is difficult to judge the significance of these differences. One may also argue that a single 200 ns MD simulation is unlikely to provide adequate sampling for statistically significant difference to be observed.

There is also insufficient methodological information to repeat these simulations, with no description of the force field and specific MD simulation details (solvent model, size of water box, cut-offs, equilibration method, etc) provided. I also was not able to find any setup and simulation data at <https://github.com/ClaraTSchoeder/schoederlab>. All files appear to be 7 months old.

Version 2:

Reviewer comments:

Reviewer #1

(Remarks to the Author)

Gulder and team have presented a report that will be incredibly valuable for the field of biocatalysis (and beyond). In addition to sufficiently addressing my comments from the last round of review, I believe they have now gone above and beyond to improve the manuscript and I highly recommend its publication in Nature Communications! Once again, congratulations to Gulder and her team members on an impressive advance in our knowledge of VHPO mechanism and potential application in chemical synthesis!

Reviewer #4

(Remarks to the Author)

The authors have adequately addressed my previous comments.

Reviewer 1:

Thanks a lot for the nice words concerning our manuscript. We are happy you are as excited as we are about the results of our study. Below you will find the answers to your questions and comments:

- *"1) While I certainly agree that the R425 mutation is very important to have identified and provides insight into the enzyme having a sophisticated way of creating an organic substrate binding site, I think it's critical that the authors spend some more time framing what these results mean (or could mean) for wild type VHPO enzymes...."*

Thanks a lot for this interesting and valuable comment. By comparing the structures of different wild-type VHPOs with the structure of our *AmVHPO*-R425 mutant, especially with regard to the exposure of the active site, we intended to provide a first preliminary correlation in the manuscript and the SI (chapter 9). During the revisions, we have now produced mutants of the VBPOs of *Corallina pilulifera* (*CpVHPO*-R397S) and *Corallina officinalis* (*CoVHPO*-R396S) by exchanging the conserved Arg at i-3 to the active site Lys (i) and examined them for their halogen specificity. In both cases, a significantly increased chlorination activity was achieved by introducing a serine (see SI, chapter 10). This indicates that the results we obtained for the *AmVHPO* are, in principle, generally applicable and that we have thus found a widely applicable tool to design VHPOs with regard to their specificity. The results of these studies have been included in the SI (chapter 10) and in the manuscript: *"Exchanging the highly conserved Arg (position 425 in AmVHPO) in the VBPOs from C. officinalis (position 396), and C. pulifera (position 397) by a Ser resulted likewise in a significant chlorination reactivity turning them in VCPOs."*

In addition, we added a new Figure 6 to the manuscript correlating the exposure of the active site in VHPOs with their halogenation specificity. More detailed studies on the structure of these new mutants and an extension of this principle to VHPOs beyond *AmVHPO*, *CpVHPO*, and *CoVHPO* are currently the subjects of research projects in our group and outside the scope of this manuscript.

In addition, we rephrased the end of the conclusion and outlook section that hints at the effects of our findings on future work using VHPOs. *"Overall, the mechanistic insights presented in our study advance the general understanding of oxidative halogenations in VHPOs. This gives us a handle for easily classifying and manipulating wild-type VHPOs and tailoring them to the needs of modern organic chemistry. Given the advantages of VHPOs for organic synthesis compared to classical chemical catalysts, especially regarding sustainability and environmental protection, we are convinced that our findings will lay the foundation for their further engineering and, thus, biotechnological application in the future."*

- *"Was x-ray crystallography attempted with the wild type enzyme with the TMB substrate? What confirmation is there that TMB doesn't bind? It would be helpful to know that this was attempted, and the crystal structure did not have TMB incorporated, for example."*

and

- *“Have any of these structures been collected with bromide bound AND organic substrate bound? I ask this based on the proposal by Littlechild (Journal of Inorganic Biochemistry, 2009, 103, 617-621.) where it is suggested that bromide binding leads to the formation of a hydrophobic tunnel that facilitates organic substrate binding.”*

We thank the reviewer for this comment. Within our previous study on AmVHPO (*ChembioChem*, **2016**), we tried to co-crystallize the wild-type AmVHPO with TMB, MCD, bromide, chloride, TMB and bromide, TMB and chloride, and MCD with bromide. Unfortunately, none of these experiments were successful. Attempts to crystallize the R425S mutant with chloride or bromide did not lead to crystals suitable for analysis by X-ray crystallography, at least in our hands.

- *“If it’s been established that the wild type AmVHPO can brominate TMB (Chembiochem, 17, 2028-2032), then I’m trying to understand how the authors think this is occurring. Presumably it is through a binding event followed by bromination, but how would it do this if the TMB doesn’t have a long enough residence time? I think the authors do allude to some “on the surface” proposals, but having a more concrete section on this would be informative for the reader.”*

We thank the reviewer for this interesting comment. If no substrate is present inside the enzyme, the electrophilic brominating species, most probably in the form of HOBr, will diffuse out of the active site. There, it can directly react with TMB in bulk solution or with an amino acid sidechain from the enzyme surface. Depending on the amino acid, a less reactive, e.g., NHBr or C=OO-Br species, can be formed, which can then act as the halogenating reagent. In wild-type AmVHPO, the active site is open to the surface with no binding site present, this scenario is very likely to take place. As demonstrated in our prior work, control experiments without enzymes but chemically generated HOBr present in our reaction mixture did not lead to similar results. Therefore, we and other groups, such as the Hartung group (Kaiserslautern), hypothesize the latter pathway. As this is purely speculative and not part of this study, we would prefer not further to elude on this aspect in this manuscript. Investigations in this direction are underway in our laboratory.

- *“I’d like to request that the authors include control runs (no enzyme, no vanadate salt, no halide, no hydrogen peroxide) for their investigations in the supporting information.”*

We thank reviewer 1 for this suggestion and added the missing information to the SI (chapter 3.1).

- *“Line 152 in the manuscript has hydrogen peroxide listed after “(see SI, chapter 5)”. Was this intentional?”*

We thank the reviewer for spotting this mistake. We moved the note “(see SI, chapter 5)” to the end of the sentence.

- *“Despite it not being the focus of this study, I would suggest adding a figure with the other substrates tested in the core manuscript rather than the SI. (discussed in lines 177-184 in the*

manuscript). As part of this, the addition of what the results look like with the wild type enzyme would be interesting for direct comparison."

We initially put the tested, very preliminary substrate scope to the SI in order to avoid the reader's attention being drawn to the development of a biocatalytic halogen method instead of the mechanistic studies. As correctly stated by the reviewer, this manuscript's main focus is mechanistic studies. Given the reviewer's suggestion, we thought about this again and added the substrates as examples to Figure 5 (Figure 5d) in the manuscript and described them briefly in the text to give the reader an impression of what might be possible with our mutant. Although the shown examples did neither arise from a comprehensive survey of substrates nor did we optimize the reaction conditions to achieve the best yields possible, they exhibit interesting information on reactivity and selectivity (see text below), corroborating the formation of a tunnel structure with a defined space for hosting the substrate. We added the following section to the manuscript:

*"A small range of other aromatic substrates was successfully transformed to the chlorinated products **10-15** employing the R425S mutant (Figure 5d), showing that the observed chlorination activity introduced by the single-point mutation is not restricted to **7**. Only chlorinated TMB **8** was formed using the wild-type AmVHPO albeit in very low quantities (<10%). Larger molecules, however, such as carbazoles or stilbenes (cf. SI, chapters 11.9 and 11.10), were not accepted by R425S-AmVHPO but served as substrates for wild-type AmVHPO catalyzed brominations.^{6d,15a} Higher chemical yields were achieved in the enzymatic chlorinations (except for **15**) when compared to those accomplished using chemical reagents (e.g., NCS with 1,4-diazabicyclo[2.2.2]octane (DABCO)). In addition, excellent regioselectivities were detected for the enzymatic transformations, which significantly exceeded the intrinsic selectivities, further corroborating the existence of a substrate binding site in R425S-AmVHPO."*

- *"I was curious if the authors looked for any regioselectivity trends with the R425 mutants on 1,3-dimethoxybenzene to see if it plays any role in modulating regioselectivity."*

Thanks for bringing up this interesting point! Actually, the *p,o*-chloro 1,3-dimethoxybenzene **11** is furnished (almost) exclusively (99:1 and 100:0) by both the enzymatic and the organic synthetic pathway, but we see a significant modulation of the regioselectivity for thymol and *N,N*-dimethyl aniline. Here the regioselectivity is enhanced in the enzymatic pathway from 55:45 (NCS, DABCO) to 100:0 for Cl-thymol (*p*-**10**) and 62:38 (NCS, DABCO) to 87:13 for the aniline derivative *o*-**12**. We added this information to the manuscript (Figure 5d) and in the manuscript and to chapter 11 in the SI.

"In addition, excellent regioselectivities were detected for the enzymatic transformations, which significantly exceeded the intrinsic selectivities, further corroborating the existence of a substrate binding site in R425S-AmVHPO."

- *"Despite the encouraging results with the TMB vs MCD competition experiments, is there literature that could be referenced that looks at the relative halogenation rates of TMB (or other substrates that have been used in the same type of experiment) with a standard halogenating agent like *N*-bromo/chlorosuccinimide? My hunch is that all these substrates might just be better inherent nucleophiles than MCD and that these competition experiments aren't as suggestive as the*

community anticipates they are. In this case, I would suggest either finding that data or just running the comparative halogenation reaction between TMB and MCD with NBS/NCS to ensure the trend you're seeing isn't just differences in nucleophilicity of the substrates. I should mention that I liked this competition experiment, but this would be just to provide a little more insight into how fair this competition is."

We thank the reviewer for this thoughtful comment. Unfortunately, we could not find any literature data. In particular, aromatic chlorinations are often tough to achieve. For the chlorination of TMB and the other aromatic substrates reported in this manuscript, we needed to activate NCS by the Lewis base DABCO to achieve chlorination, while MCD was converted, albeit slowly, without DABCO. Therefore, we assume our TMB vs MCD competition experiments point in the right direction and are not due to differences in nucleophilicity.

Reviewer 2:

- *"No evidence, other than visual inspection of enzyme surface structure, is given to support this and thus further supporting evidence with additional chlorinating V-HPOs would be required to warrant publication in Nature communications."*

The manuscript reports on the elucidation of the molecular basis responsible for halogen and substrate specificity in VHPOs using the *AmVHPO* as a model enzyme. The conclusions drawn by the authors are based on in-depth studies using common tools in structural biology, molecular biology, biochemistry, biocatalysis, chemistry, computational science, and machine learning and not only "visual inspections of enzyme surface structures", as stated by this reviewer. In our opinion, putting our findings in a broader context, particularly what this could mean for other wild-type VHPOs, is highly warranted for a manuscript in a high-class and, thus, high-impact journal. This is in agreement with reviewer 1's opinion, who requests an even more intense discussion of this aspect. To underline our arguments, we compared the crystal structures of different wild-type VHPOs that are available in databases and their reported halogen specificity with those of the wild-type and R425S-*AmVHPOs*.

To further corroborate our statements regarding the general applicability of our finding, we now mutated the conserved Arg in the VBPOs of *Corallina pilulifera* and *Corallina officinalis* and examined the mutants for their halogen specificity. In both cases, chlorination activity increased significantly by introducing a serine at the same position as in the VHPO from *A. marina* (see SI, chapter 10). This indicates that the effects we obtained for the *AmVHPO* are, in principle, generally applicable and that we have thus found a tool to design VHPOs with regard to their specificity. The results of these studies have been included in the SI (chapter 10) and in the manuscript: *"Exchanging the highly conserved Arg (position 425 in AmVHPO) in the VBPOs from C. officinalis (position 396), and C. pulifera (position 397) by a Ser resulted likewise in a significant chlorination reactivity turning them in VCPOs."*

More detailed studies on the structure of these new mutants and an extension of this principle to other VHPOs are currently the subjects of research projects in our group. These studies constitute **years of additional work and several PhD theses** and are thus **outside the scope** of this manuscript and **outside a "communication"** article in general.

- *"Please use either the three letter amino acid code or one letter code consistently throughout the manuscript."*

We used the three-letter amino acids code when we described actual amino acids or their side chains and their functions in the text as this is, in our opinion, easier to understand for the broad readership of Nature Communications, which might not all be that familiar with the less intuitive one-letter code. The one-letter code was used when we described genetic modifications to the amino acid sequence as mutants, etc. This format is commonly used in articles and, in our opinion, makes the article easier to read.

- *"Line 44 – 46 "Although recent achievements in homogeneous catalyst development have led to more selective catalytic halogenation approaches, these catalysts tend to be intricate, expensive, and time-consuming to synthesize." – references should be provided"*

We added reference 3 to the manuscript

- *"Line 48 – 50 "The most common strategy in nature is electrophilic halogenation facilitated by vanadium- (VHPOs) and heme-dependent haloperoxidases, and flavin-dependent halogenases." – references should be provided"*

Here, we already cited the appropriate literature (now ref. 5). The 3 enzyme classes named in the text are the only ones capable of electrophilic halogenations. As this is textbook knowledge, we do not see why an additional citation is needed.

- *and it should be mentioned that heme-dependent haloperoxidases have a similar mechanism to VHPOsx"*

We disagree with the reviewer that VHPOs and heme-dependent haloperoxidases have a similar mechanism. While heme-dependent enzymes produce the electrophilic halogen species via oxidation of the Fe cofactor, the vanadium cofactor is redox neutral in VHPOs. Only the electrophilic aromatic substitution is common in both enzyme classes, but this is also the halogenation mechanism operative in flavin-dependent enzymes and depends on the electrophilic X reagent and not on the enzyme.

- *"Line 95-97 "Mutagenesis experiments in the second shell combined with activity screening assays revealed that R425S-VCPO was transformed into a chloroperoxidase capable of efficient aromatic chlorination, even on a preparative scale." This suggests that there was no chlorination activity for the wild type enzyme. No mention of isolated yield for preparative scale experiments"*

We are very sorry, but we do not understand why our statement suggests that there was no chlorination activity in wild-type *AmVHPO*. The wild-type *AmVHPO* shows only very weak chlorination activity, which is restricted almost to TMB as shown by our experiments in this manuscript and SI (<10%, 72h) and our previous studies, resulting in none "efficient aromatic chlorination". Further

substrates tested by our group were not chlorinated by the wild-type AmVHPO (see Figure 5d and SI) at all. This is the reason why the wild-type AmVHPO is a VBPO. Our mutant triggers "efficient aromatic chlorination," as shown in Figure 5d and the SI. Therefore, we call the R425S mutant a VCPO. To make clear that our preliminary substrate scope shown in this article relies on isolated and not on LC or GC yields, we added the following note to the Figure Caption of Figure 5 in the manuscript: "^aisolated yield using R425S-AmVHPO as catalyst; ^byields determined by LC-MS or GC-MS using wild-type AmVHPO as catalyst; ^cisolated yield using NCS/DABCO; n.r. = no reaction; r.r. ratio of regioisomers."

- "Lines 122, 161, 292 Naming position 425 as "i-3" is confusing and unnecessary. It also sometimes becomes "i+3".

We used this annotation as a rapid and intuitive way to show the distance to the active site lysine (position 428, i), as defined in the manuscript. It also makes it more comparable to the amino acid sequences of other enzymes, in which the numbering of the amino acids must not be identical. As it seems to be too confusing for the reviewer, we changed the description of the amino acid positions in the text accordingly.

"...residue is located three amino acid residues apart from the active site lysine..." and

"...the change of Arg425 did not affect the binding..."

- Line 141 – 145 Data confusing and contradictory. SI Sfigure 3 shows no activity for the chlorination of monochlorination with wild type, here authors state 8% chlorination of wild type. Errors should be included in figure 3 data."

We do not see how this data is confusing or even contradictory. The data reviewer 2 is referring to originates from different measurements from differently produced enzymes. The results in Figure 3a manuscript describe the MCD bromination and chlorination assays directly from our mutagenesis experiments (from 96-well plates) using the cell lysate (see chapter 2, SI). SFigure 3 in the SI describes the activity of purified enzyme after heterologous expression in *E. coli* on a larger scale (chapter 3, SI). We changed our statement below in SFigure 3 to "Wild-type (black) and R425D variant (blue) showed almost no activity in the chlorination of monochlorodimedone,..." as the graph is not straight but very weakly declining over 2000s. We added the error bars to Figure 3a and SFigure 3, SFigure 5, and SFigure 6.

- "Line 163-165 "As different amino acids three positions apart from the active site lysine (Lys428) trigger an enhanced chlorination activity in different VHPOs, such as, e.g., Phe and Trp in CpVHPO, Trp in AnVHPO or Ser in AmVHPO"- References need to be provided, has the "enhanced chlorination" be probed previously?"

We repeated the citation of reference 17 and the PDB number of the respective enzyme crystal structures at this position in the manuscript.

- *"Fig. 3. A) The serine variant stands out with a ten times higher conversion of MCD (3) than the wild-type." - Needs to be made clear that this is only for chlorination, not bromination."*

We changed the sentence accordingly to *"The serine variant stands out with a ten times higher conversion of MCD (3) than the wild-type for the chlorination reaction."*

- *"Fig. 3. D) – Not convincing data, why does conversion decrease between time points? Error bars need to be made much clearer"*

Error bars are actually included in both graphs in Figure 3D but are very small. We do not know why we have this very small drop in TMB conversion between 15 min (9%) and 30 min (5%) and then a steady increase in TMB conversion. One reason might be solubility effects, but as this is purely speculative, we will not include this in the manuscript. We repeated this reaction several times and detected always the same curve progression. In the mutant, a plateau in TMB conversion can be seen between the time points 30 min and 1h, which might arise from the same phenomenon. Overall, the authors do not see why this data is not convincing.

- *"Line 181 – 184 "Intriguingly, a range of other aromatic substrates, such as TMB (7), thymol indoles, pyridines. were transformed to the corresponding chlorinated products with excellent regioselectivities and high yields (see SI, chapter 10)." - Actually a very small range of substrates, only 5, with ranges of conversion between 24 - 95%, 2 show regioisomers formed. Would not call this excellent regioselectivity or high yields. Indeed no isolated yields are reported."*

We initially put the tested, very preliminary substrate scope to the SI in order to avoid the reader's attention being drawn to the development of a biocatalytic halogen method instead of the mechanistic studies. As correctly stated by reviewer 1, this manuscript's main focus is mechanistic studies. Nevertheless, we now added more substrates and the isolated yields to the manuscript. In addition, we compared the chlorination reactivity and the regioselectivity achieved by our enzyme with those observed with standard organic reagents (see detailed answer to reviewer 1's comment). The isolated products have been characterized by NMR and mass spectrometry and the data has been added to the SI, chapter 11.

- *"Line 221 – 223 "However, for the highly reactive hypochlorous species HOCl, the chlorination reaction likely occurs immediately after the in-situ formation of the electrophilic chlorine reagent, rendering the halogenation of an enzyme-bound substrate very likely"- based on what? Reference for the comparative reactivity of HOCl and HOBr?"*

As this sentence did confuse reviewer 2, we deleted it from the manuscript.

- *"Line 225 – 226 "The hypothesis of a substrate binding site in R425S-AmVHPOs was thus further verified." – Overreaching statement"*

The sentence was changed to *"showed a preference for TMB over 3 (see SI, chapter 6), thus supporting our hypothesis of a substrate binding site in R425S-AmVHPOs."*

- *"Fig. 5. - This figure is not effective at showing that TMB only binds to the R425S mutant. Why are superpositions of the liganded R425S with wt and R425D included? Currently it just looks like the substrate is bound in every enzyme. There should be a clear figure showing that only R425S binds the substrate in this way."*

The intention of this figure was to show that the region where the substrate TMB binds in the mutant is only defined in the R425S mutant and not in the non-chlorinating enzymes, wild-type AmVHPO and the R425D mutant. To make clear that **7** only binds to the R425S mutant, we stated in the manuscript that *"...TMB (**7**) **only binds to** the AmVHPO once the ordered loop region at the interface of two enzyme subunits (loop region, residue 390-404, cyan..."* and in the caption of former Figure 5 *"TMB (**7**) is **only bound in the R425S mutant** (PDB ID 8Q22). Superposition of the liganded AmVHPO-R425S mutant with wild-type (PDB ID 5LPC) and R425D (PDB ID 8Q20) structures illustrates **how 7 may bind** in these variants."* As this might confuse other readers in a similar way as reviewer 2, we added the illustration of the R425S mutant to Figure 4 (now Figure 4c) and moved Figure 5 to the SI.

In addition, we added our results from the MD calculations to this section to further corroborate our results shown in Figure 5. Therefore, we changed the text in the following way: *"TMB (**7**) only binds to the AmVHPO once the ordered loop region at the interface of two enzyme subunits (loop region, residue 390-404, cyan, Fig. 4c) is defined. In the R425D mutant, this motif is only partially present (residue 390-400), so no interactions with **7** can occur. These molecular findings agree with the wild-type structure, in which residues 390-404 are also flexible. Thus, the plasticity of the specificity pocket depends on the introduced mutant, and only a small modular sequence motif coordinates substrate selection (Fig. 6a). No co-crystal structure of **7** and the R425D or the wild-type enzyme could be obtained. The effect of the R425D mutation on loop flexibility and pocket formation was further confirmed by short Hamiltonian replica-exchange molecular dynamics simulations. In these simulations, the loop showed a distinctly different conformational profile (see SI, SFig. 19A) in the wild-type and R425S mutant. In agreement with the crystal structure, we observed higher flexibility for several positions in the loop of the wild-type. In particular, position Phe401 stood out as less stabilized as compared to the wild-type protein (see SI, SFig. 19B). As this residue is close to the active site and identified as part of the binding pocket, a certain set of side-chain orientations may be required to allow TMB access and binding (see SI, SFig. 19D). This may indicate a conformationally dependent mode of substrate access or binding."*

- *"Line 265 -268 - "A significant decrease in chlorination activity was indeed observed in the MCD (3, conversion of 3 was decreased by 20% after 15 min) and in the phenol red assay. At the same time, no bromination was detectable at all. Moreover, the chlorination of TMB (7) was only detectable in traces according to GC analysis" – This is confusing and needs to be simplified, essentially E139G reduces activity for both chlorination and bromination?"*

We are sorry but we could not see how these statements are confusing. For the chlorination by the R425S,E139G double mutant, the chlorination is decreased (but still occurs), and for the bromination, it does not occur at all. In order to attribute to the concerns of reviewer 2, we revised the section to *"The E139G variant, however, showed a different behavior. A significant decrease in the*

chlorination rate of MCD (**3**, conversion of **3** was decreased by 20% after 15 min) and phenol red assay (**5**) was observed (see SI, chapter 4). Moreover, the chlorination of TMB (**7**) was only detectable in traces according to GC analysis. This emphasizes the supporting function of the carboxylic acid for delivering the electrophilic halide species HOX to the substrate and its role in stabilizing the cationic Wheland intermediate **9** (Figure 5c)."

- "Line 270 Wheland spelling is incorrect"

We thank the reviewer for spotting this typo and changed it accordingly.

- "Line 302 – 305 "the Ser425 from the adjacent subunit alters the hydrogen-bonding network within the active site, thus forming a tunnel protecting the vanadate in the active site. This reduces the access of solvent molecules to the vanadate cofactor and thus abolishes the degradation of the reactive Cl⁺ species"- This appears to be speculative and further evidence should be provided, minimally a model"

The formation of the tunnel and the vanadate/phosphate being more protected as in the wild-type enzyme and the R425D mutant can be easily seen from the X-ray crystal structures reported in the presented manuscript and the work published earlier by our group. That within a defined structure, such as a tunnel, the access of molecules, such as solvent molecules, is hindered is, in our opinion, a logical conclusion. The same holds true for the degradation of a reactive species if this species is offered to other molecules. Therefore, these statements are not speculative at all. Actually, similar explanations are commonly used in the literature concerning flavin-dependent halogenases, where tunnel formation is also a key feature. Nevertheless, attributing to the concerns of reviewer 2 we changed the sentence toward "This might reduce the access of solvent molecules to the vanadate cofactor and thus most likely abolish the degradation of the reactive Cl⁺ species."

Studies on this behavior are indeed highly interesting but, again, are beyond the already rich scope of results presented in this communication.

- "Supplementary Information Chapter 6- Competition assays, S11, Showing that as increasing amounts of TMB is added to the reaction the consumption of MCD is slowed- how does this show that there is any TMB binding rather than just showing the equivalents of HOCl are getting used up by added TMB?"

We are sorry, but we do not understand in which direction reviewer 2 is pointing. The curves show nicely that TMB conversion is faster than MCD conversion. Both substrates use HOCl as chlorinating reagents. Please refer to the comment of reviewer 1 and our answer for further details on this competition assay.

- "Chapter 9- Comparison of structure, It would be good to include some of these figures in the main text, as they are the main argument for active site accessibility controlling chlorination activity."

We strongly disagree with reviewer 2 that the comparisons of the X-ray crystal structures of various VBPOs and VCPOs are the **main** argument for the active site accessibility controlling chlorination activity! On the contrary, this is more of a first step in putting the results we obtained for our model enzyme *AmVHPO* in a broader context, as even further suggested by Reviewer 1. Elucidating the mechanism of halogenations catalyzed by VHPOs is not trivial and has been the subject of numerous important investigations within the last decades. Despite the landmark achievements in this field by many other groups, many questions still remain unanswered. With our study presented in this manuscript, we combined computations, biocatalysis, molecular biology, directed evolution, and structural biology. Only by applying an interdisciplinary approach, as we did, it becomes possible to achieve another, in our eyes, significant step forward, revealing the complex catalytic process within VHPOs. To address this point further, we added a new Figure 6 to the manuscript showing the different degrees of exposure of the prosthetic group in the VHPO's active sites to the surface.

- *"Chapter 10- Aromatic Chlorination, Protocol does not make clear if halogenated products have been extracted/purified, what the isolated yields were, or what the NMR analysis is of. Presumably the crude extract? Copies of the NMR trace should be included and references to literature data. There appears to multiple compound numbers for the same compounds? Many LCMS traces have multiple uncharacterised peaks. Are these side products? What are they? All data should be presented in standard formats."*

As mentioned before, we extended the biocatalytic transformation of aryl compounds by using the R425S mutant as the catalyst.

Reviewer 3:

- *"Furthermore, the observation of the loop ordering per se is not sufficient to explain the halide reactivity without other studies, such as MD simulations."*

We thank this reviewer for this comment. We conducted MD simulations on the loop structuring. These results have been included in our manuscript and in the SI.

- *"Fig. 1: In panel a) the oxygens and the hydrogens need to be balanced properly on the right- and the left-hand sides, so please include additional product species. Somewhat related to this issue, this chemical scheme needs to be redrawn in a sequential way. The formation of HOX is in the second reaction step, not in the reverse step of the first reaction (if the reaction was reversible as drawn, H2O2 would be the product of the reverse step)."*

As this is a "schematic" drawing of the reaction and not an exact reaction equation a proper balance of the atoms is not needed. In addition, we have not drawn the arrows stating reversibility (equilibrium). Here, we used two full-headed arrows and not two arrows with only half-arrow heads as used for reversible transformation. To make it more clear, we can use curved arrows to show further emphasize that we are dealing here with a catalytic cycle and not a reversible reaction.

- *Fig. 1 needs to be conventionally formatted, without panel headings and embedded questions. Panel d) looks like an online abstract, rather than a figure panel. This figure additionally needs a major revision: the structure and its elements are not well depicted, and they may need to be shown as a cartoon with zoomed in regions as insets (instead of the text there), with judicious use of color. This is also an issue for Fig. 4. The color labels will likely need to be removed from the figure legend and described in words.*

We thank the reviewer for commenting on the style of our Figures. However, this issue will be discussed with the editor and the production team in case the manuscript is accepted. Then we will gratefully consider the points raised by this reviewer.

- *Lines 91-108 are largely a repeat of the Abstract. Normally sentence in line 90 should conclude the Introduction.*

We thank reviewer 3 for pointing out this issue. To my knowledge and my experience publishing in Nature journals, a short summary of the content of the manuscript is required at the end of the introduction. Therefore, we added this part to the manuscript which we would not do for other journals. Here again, I would like to leave the decision to the editor.

- *Line 113-114 and then again in line 119. Need a reference to structures. Because you used the experimental structure as a starting point, you need to state so in line 119.*

The reference to all these points is ref. 6d which is cited in line 112 (now l. 117) and line 117 (l. 122). To follow the suggestions by the reviewer we are re-citing 6d in the mentioned lines (now l. 119 and l. 124) again.

- *Line 120: what is meant by "predicted to respond well". What is the response being observed? Similarly, in line 129, what is meant by "energetic advantage"? Is this borne out experimentally?*

We thank reviewer 3 for this comment. We added the filters used to design the sequence by ProteinMPNN and referred to the sections in the SI where the details of these studies can be found.

- *Line 143: you are referring to relative conversion- this is not the reaction rate.*

We thank the reviewer for pointing out this issue. We changed the sentence accordingly: "*Strikingly, the conversion of 3 by bromination and chlorination drastically increased...*"

- *Line 67: reword "low halogenation reactivity of HOX in solution" to "low halogenation activity and chemical instability of HOX in bulk solvent"*

&

- *Line 69, Fig. 1 legend and elsewhere: use "superimposition" instead of "overlay" or "alignment", when describing structure;*

We thank the reviewer for bringing up this issue and changed it accordingly.

- *Line 76: please use a concrete scientific term instead of "oxidizing power", i.e. catalytic efficiency, catalytic activity, substrate scope, etc, as applicable here.*

"As none of the terms suggested by reviewer 3 fits in the sentence, we changed it to *"Nevertheless, VHPOs differ in their capability of oxidizing different halogens."*

- *Line 77: the term "outer sphere" needs to be explained structurally. Or is this term just used to mean residues that are not conserved, but are in the vicinity of the catalytic residues? Please clarify.*

The term "outer sphere" refers to the amino acids which are not present within the active site. For VHPOs, the active site refers to the amino acids binding the prosthetic group. The term outer sphere or 2nd / 3rd sphere is frequently used in VHPO literature.

- *Line 99: did the authors mean "intrinsically disordered"?*

We are not sure what the reviewer refers to as line 99 talks about intrinsically distorted.

- *Fig. 3: absorbance, not absorption*

We thank the reviewer for bringing up this issue and changed it accordingly.

Reviewer 1:

Thanks a lot for the nice words concerning our manuscript. We are happy you are as excited as we are about the results of our study. Indeed, we agree that our study, together with the great work of other groups, is just a start, although an important one, to shed light on the complex processes occurring during VHPO-catalyzed halogenations.

Reviewer 2:

- *“...Unfortunately, the co-crystallization of the substrate 7 with the protein was not fruitful. Thus, the key experimental data currently supporting this hypothesis are the results from the competition assay...”*

We do not understand the comment by reviewer 2 that the co-crystallization of TMB (**7**) was not fruitful in the *AmVHPO*-R425S mutant. There is a significant change in the electron density of the X-ray crystal structure of *AmVHPO*-R425S with TMB (PDB ID 8Q22) when compared to its apo form (see Supplementary Data, chapter 8.3 and 8.4), showing the binding of the substrate. The experimental data obtained from the X-ray crystallographic analysis of the different *AmVHPO* crystal structures provide the basis for our hypothesis. The competition assay corroborates the binding of TMB in the mutant. In addition, we observe a modulation of chemo- and regioselectivity for the chlorination catalyzed by *AmVHPO*-R425S. While the wild-type *AmVHPO* is capable of brominating aromatic compounds independent of their molecular size, *AmVHPO*-R425S cannot chlorinate larger aromatic compounds, such as carbazole (**S8**) and 3,5,4'-trimethoxystilbene (**S10**), albeit both substrates show similar electronic properties as the substrates which were successfully chlorinated (see Supplementary Information, chapter 11). In addition, we see a significant modulation of the regioselectivity for thymol and *N,N*-dimethyl aniline. Here, the regioselectivity is enhanced in the enzymatic pathway from 55:45 (NCS, DABCO) to 100:0 for Cl-thymol (*p*-**10**) and 62:38 (NCS, DABCO) to 87:13 for the aniline derivative *o*-**12**. This information has been added to the manuscript (Figure 5d) and chapter 11 in the SI already in the previous version. This selectivity change, compared to the expected intrinsic selectivity, further hints at the chlorination taking place within the enzyme and not outside and thus for a substrate binding site playing a significant role during the chlorination event. This observation is also in line with the comment by Reviewer 1: *“I also found it very interesting during this revision that the R425S variant of AmVBPO for N,N-dimethyl aniline and thymol. To me this provides indisputable evidence for a tunable organic binding site that will be very exciting for the biocatalysis/organic synthesis community!”*

- *“..., the authors would need to improve the clarity of the kinetic data on substrates 3 and 7.”*

As suggested by reviewer 2, we improved the clarity of the data presentation in the competition assay. Therefore, we plotted the initial velocities of MCD (**3**) against the excess concentrations of TMB (**7**) over MCD (**3**) for the wild-type *AmVHPO*-catalyzed bromination, which should occur outside the

enzyme by freely diffusing HOBr and substrate, and the AmVHPO-R425S-catalyzed chlorination (see Supplementary Data, chapter 8), which should occur for TMB (7) inside and, according to literature, for MCD (3) outside the enzyme. The results and their explanation, as shown below, were added to the SI:

“SFigure 14 | Competition of MCD (3) consumption with increasing concentrations of TMB (7). Plot of initial velocities of MCD (3) consumption against the excess of TMB (7) over MCD (3). A: wild-type AmVHPO in the presence of bromide. B: R425S-AmVHPO in the presence of chloride. Assays were performed in triplicates.

SFigure 14 shows the kinetic analysis of the competition assays between MCD (3) bromination and chlorination. Shown are the initial velocities of MCD (3) halogenation plotted against the excess of the concentrations of TMB (7) over MCD (3). Both curves follow an exponential decay described by $v = Ae^{-x/\tau} + v_0$. v_0 . This describes the velocity the system evolves to at an ever-increasing excess of TMB (7). τ describes the excess of TMB (7) at which the system reaches a velocity of $1/e$ (appr. 37%) of the highest velocity. For the graph in SFigure 14A, which shows bromination in the presence of wild-type AmVHPO, v_0 eventually approaches 0 with τ at approximately 2, meaning 37% of the starting velocity is reached at a twofold excess of TMB (7) over MCD (3).

In contrast, in SFigure 14B, where chlorination in the presence of R425S-AmVHPO is depicted, the graph plateaus already at a higher level, taking into consideration that chlorination generally proceeds much slower than bromination and initial velocities are therefore lower. Furthermore, τ is at approximately 1.3, meaning TMB (7) reduces the initial velocity to 37%, already at a 1.3fold excess over MCD (3).

These results lead to two conclusions: The lower τ value in SFigure 14B hints that TMB (7) is converted to the chlorinated species quicker compared to the bromination in the wild-type enzyme. If conversion to the chlorinated species is only dependent on the presence of a free-diffusing HOX species interacting with free-diffusing substrates, one would assume that τ is comparable to or higher than the value for bromination, as TMB (7) is less reactive than MCD (3). However, the lower τ can be attributed to the fact that TMB (7) is binding to the mutant enzyme and blocks chlorination of MCD (3) already at lower concentrations. Another interesting observation is that TMB (7) cannot reduce the velocity of MCD (3) chlorination to zero as opposed to bromination in the wild-type enzyme, meaning that there is a saturation of the effect starting at an approximately 2fold excess of TMB (7) over MCD (3). Above that value, we do not see a deceleration of MCD (3) chlorination, meaning that free-floating, non-binding TMB (7) is not converted to the chlorinated species. This underlines our hypothesis that chlorination of TMB (7) can only happen in close proximity to the active site of the enzyme. A substrate binding pocket, as we see it in our crystal structure, seems to be a plausible explanation. As TMB (7) most probably does not bind with high affinity as it is a small molecule with only one aromatic moiety, it is not surprising that chlorination of MCD (3) cannot be completely inhibited in such a case.”

Reviewer 3:

We thank reviewer 3 for his comments that helped to improve the manuscript.

Concerning reviewer 3's editorial comments from the first revision that have not been adequately addressed:

- *Fig. 1 needs to be conventionally formatted, without panel headings and embedded questions. Panel d) looks like an online abstract, rather than a figure panel. This figure additionally needs a major revision: the structure and its elements are not well depicted, and they may need to be shown as a cartoon with zoomed in regions as insets (instead of the text there), with judicious use of color. This is also an issue for Fig. 4. The color labels will likely need to be removed from the figure legend and described in words.*

We thank the reviewer for commenting on the style of our Figures. However, this issue will be discussed with the editor and the production team in case the manuscript is accepted. Then, we will gratefully consider the points raised by this reviewer.

- *Lines 91-108 are largely a repeat of the Abstract. Normally sentence in line 90 should conclude the Introduction.*

We thank reviewer 3 for pointing out this issue. To my knowledge and my experience publishing in Nature journals, a short summary of the content of the manuscript is required at the end of the introduction. Therefore, we added this part to the manuscript which we would not do for other journals. Here again, I would like to leave the decision to the editor.

We leave it to the editorial office if specific changes should be made. Nevertheless, we changed the formatting of Fig. 1 without changing its content and hope it now better fits the taste of reviewer 3.

Reviewer 4:

- *“However, the wt simulation is built on a model of the 425S variant, so without evidence that the wt simulation is adequately equilibrated, it is difficult to judge the significance of these differences. One may also argue that a single 200 ns MD simulation is unlikely to provide adequate sampling for statistically significant difference to be observed. There is also insufficient methodological information to repeat these simulations, with no description of the force field and specific MD simulation details (solvent model, size of water box, cut-offs, equilibration method, etc) provided. I also was not able to find any setup and simulation data at <https://github.com/ClaraTSchoeder/schoederlab>.”*

Thank you for your detailed evaluation and valuable feedback. We apologize that the data provided on GitHub was not accessible. We uploaded all data now on Zenodo (DOI 10.5281/zenodo.14544279), which can be accessed via the following link:

<https://zenodo.org/records/14544280?preview=1&token=eyJhbGciOiJIUzUxMiJ9.eyJpZCI6ImZlYXN0IiwiaWF0IjoiMj02MDUyMjM0MmQ4ZjE1ZCJ9.4i8nlmdKbwJ7yg6azw0twosL289lzJ3aG-R7E7kbZ1q4qsXeMI8zRE7JK1mZ4x1c2Qyc093sfXaN2szcSCq5Mw>

This is currently a preview. We will publish the data immediately after acceptance of the manuscript. We agree that clarifying our methodology is necessary to understand and reproduce our simulation results. The method section lacked details, leading to wrong assumptions about the performed

protocol. We extensively edited the description of our protocol to allow for the reproduction of our simulations. In addition, we made sure that the mdp files for the equilibration and the production run are publicly available on Zenodo, as well as the necessary topology and trajectory files.

To address sampling adequacy, we noticed a mistake and clarified the performed HREX protocol. As it describes a replica exchange protocol using 8 replicas (not 5, each run for 200 ns), which exchange complete states with a given probability, the cumulative simulation time was 1.6 μ s per system. A temperature gradient of 300 K to 475 K between the replicas assured extensive sampling of possible loop conformations. We apologize for this mistake and thank the reviewer for his feedback.

We acknowledge the concern about the starting structure of the wt simulation being derived from the R425S variant model. We now provide additional information on the equilibration and access to the equilibration files.

- *“Figure S19, which is mis-cited in the main text as Figure SX”*

We thank the reviewer for pointing out this typo and changed it accordingly.